



# Intelligent prospector v1.0: geoscientific model development and prediction by sequential data acquisition planning with application to mineral exploration

John Mern[1], Jef Caers[2]

[1]Kobold Metals, USA

[2]Stanford University, Department of Geological Sciences, USA

*Correspondence to*: Jef Caers (jcaers@stanford.edu)

**Abstract.** Geoscientific models are based on geoscientific data, hence building better models, in the sense of attaining better predictions, often means acquiring additional data. In decision theory questions of what additional data is expected to best improve predictions/decisions is within the realm of value of information and Bayesian optimal survey design. However,

these approaches often evaluate the optimality of one additional data acquisition campaign at a time. In many real settings, certainly in those related to the exploration of Earth resources, possibly a large sequence of data acquisition campaigns need to be planned. Geoscientific data acquisition can be expensive and time consuming, requiring effective measurement campaign planning to optimally allocate resources. Each measurement in a data acquisition sequence has the potential to inform where best to take the following measurements, however, directly optimizing a closed-loop measurement sequence

requires solving an intractable combinatoric search problem. In this work, we formulate the sequential geoscientific data acquisition problem as a Partially Observable Markov Decision Process (POMDP). We then present methodologies to solve the sequential problem using Monte Carlo planning methods. We demonstrate the effectiveness of the proposed approach on a simple 2D synthetic exploration problem. Tests show that the proposed sequential approach is significantly more effective at reducing uncertainty than conventional methods. Although our approach is discussed in the context of mineral resource

exploration, it likely has bearing on other types of geoscientific model questions.

## 1 Introduction

As the world weans itself off fossil fuels over the next decades, new forms of energy will heavily rely on Earth materials, in particular minerals. Rare earth elements are used in a variety of clean-energy technologies (Hague et al., 2014). Fully electrifying the light-duty auto fleet requires discovering new ore deposits of critical electric vehicle (EV) materials: copper,

nickel, cobalt, and lithium (Savacool et al., 2020). Increasing the required supply of these critical minerals requires a yet unattained discovery rate of new deposits. Mineral exploration is slow, requiring extensive guidance from human experts. As



a result, the rate of new discoveries has declined over the last decades, since deposits with sections visible at the surface have mostly been discovered (Davies et al., 2021). At the same time, the demand will continue to increase, making minerals a targeted commodity subject to international conflict (National Research Council, 2008), social, and environmental concerns (Agusdinata et al., 2018). Enhancing and speeding up mineral exploration at a planet-wide scale is required. Our approach, using Artificial Intelligence for effective planning of exploration endeavors, aims to contribute to this challenge.

Mineral exploration requires making sequential decisions about what type of data to acquire, where to acquire it, and at what resolution with the goal of detecting an economically mineable deposit. In other words, mineral exploration is a sequential decision-making problem under uncertainty. These types of problems have previously been studied under several non-sequential frameworks in various areas of the geosciences. Optimizing spatial designs of experiments is a well-studied topic. McBratney et al. (1981) described a method for designing optimal sampling schemes based on the theory of regionalized variables (Matheron, 1971) by modeling spatial dependence with semi-variograms. The 1990s saw a significant debate arising in the soil sciences community (Brus & Gruijter, 1997; Van Groeningen et al., 1999; Lark, 2002, Heuvelink et al., 2006) around adaptation of geostatistics and its role in optimal survey design. Likewise, geostatistics-based optimal design of environmental monitoring has been significantly developed (De Gruijter et al., 2006; Melles et al., 2011). Geostatistical methods are often not Bayesian, which may be a disadvantage when the spatial structures (e.g., variograms) are uncertain themselves. A method for Bayesian optimal design in spatial analysis was developed by Diggle & Lophaven (2006).

Optimal placement of drill-holes for mineral exploration and mining (resource delineation) has received significant attention. Some methodologies aim to minimize the uncertainty on spatial properties through use of geostatistical algorithms that model the effect of measured data on spatial uncertainty (Pilger et al., 2001; Koppe et al., 2011; Koppe et al., 2017; Caers et al., 2022; Hall et al., 2022). Others rely on decision theoretic concepts of value of information to quantify the dollar value of gathered information to reduce uncertainty on an economic property of interest (Froyland et al., 2004; Eidsvik, & Ellefmo, 2013; Soltani-Mohammadi & Hezarkhani, 2013). Bickel et al. (2008) recognizes the sequential nature of the problem and illustrate that sequential information gathering is superior to non-sequential schemes, a concept that goes back to the 1970s (Miller, 1975).

The above methodologies evaluate the performance of a given spatial survey design, but do not address the combinatorial problem of creating optimal survey plans. In general, the number of sequences to evaluate grows exponentially with the number of surveys. For example, when planning a sequence of 10 surveys at 100 possible locations, there are more than 17 billion possible sequences that could be evaluated. Many problems will likely require more than 10 data acquisition actions to discover a mineral deposit that is economically feasible. Therefore, methodologies (like Emery et al., 2008) that use optimization in combination with geostatistics are likely intractable for many practical problems.



Sequential planning methods solve for each action in a sequence only after observing the results of each previous action. Planning is typically done in either an open-loop or closed-loop fashion. Open-loop methods solve for each action in the sequence that gives the best immediate return according to some metric, without considering how the information learned from taking that action is likely to impact future decisions. Closed-loop methods solve for actions that maximize the

expected return of all remaining actions in a sequence. Closed-loop methods tend to outperform open-loop methods, especially on tasks in which a lot of information is learned each step (Russell and Norvig, 2020: p.120-122). Closed-loop methods, however, tend to require significantly more computational effort than open-loop approaches.

Recent work has applied Bayesian optimization to develop open-loop solutions to sequential experiment design (Shahriari et

al., 2016). Marchant et al. (2014) specifically consider the application of Bayesian optimization to spatial-temporal measurement sequences. Receding horizon control has been used in sequential resource development (Grema et al., 2013) in conjunction with general particle swarm optimization. While these methods may be tractable, they are likely sub-optimal over the entire measurement sequence, since each action only optimizes its own return.

Closed loop methods solve for optimal conditional sequences of actions. Common closed-loop methods include reinforcement learning, dynamic programming, and Monte Carlo planning. These methods search for optimal actions through extensive interaction with a simulation of the target environment. Because of the large amounts of data required, these methods were initially developed on virtual domains such as video games (Chaslot et al., 2008). Recently learning-based approaches have achieved state-of-the-art performance in several real-world domains including autonomous driving

(Brechtel et al, 2014) and robotic control (Grigorescu, 2020). Little work has been done, however, in applying these approaches to resource exploration. Torrado et al. (2017) proposed a Monte Carlo planning method for a similar task of optimal sequential reservoir development. This work, to the authors knowledge, is the first proposal for a general approach to optimal closed-loop decision making for geoscientific sequential data acquisition planning.

**2 Illustration case for sequential data acquisition planning in resource exploration**

Our development will be illustrated on an analogue case set-up that contains many elements common to resource exploration planning. In that sense we aim for modularity in the development where several components (inverse modelling, geological modelling, data forward modelling) can be changed out without changing the sequential data acquisition methodology.

Specifically, we will focus on the exploration of one or more orebodies in the subsurface. The elements of the problem definition consists of 1) a description of the state of knowledge of the physical world, 2) a description of data that exists or is planned to be acquired on the physical world, 3) rewards and costs associated with the exploration endeavor.





Knowledge and uncertainty about the subsurface is commonly represented by probability distributions over the parameters
of the subsurface system. Gridded models describing parametric distributions over geological, geophysical, and geochemical
properties may be too high dimensional for practical use in decision making. A realization (in geostatistical jargon)
generated from a probability distribution over the subsurface represents a plausible representation of the physical world. An
ensemble of plausible realizations is a tractable method to represent the distribution over the subsurface. The variation
between multiple realizations is an empirical representation of uncertainty (lack of knowledge).


A subsurface orebody may be hard to identify in a real setting for various reasons. In geophysical surveys, many other
geological features may act as ore bodies. An orebody is also not necessarily a perfect anomaly in a homogenous geological
setting. Tectonic, metamorphic, sedimentary, and other alteration processes may have changed the nature of the original
orebody. In Figure 1, we show how we created an analogue situation that mimics many of these elements. Figure 1
represents a simplified 1D depiction, though the methodology will be applied to 2D and 3D settings. Figure 1 should only be
referenced as a template containing the challenges present in mineral exploration.

First, we represent the mineralization by the function in Figure 1A. The example shows a unimodal function, however a
multiple of these mineralization bumps may be present. Second, we introduce a "geological background variation" as shown
in Figure 1B. This represents all geological processes that have altered the original ore-body shape. This variation is not
entirely random and has some structure. In our setting, we model it as a Gaussian process with known correlation structure
(variogram). In practice, a much more complex model of the background geology may be used with the presented methods.
By adding the "mineralization field" to the "geological background field", we obtain the "measurable variation" shown in
Figure 1C. When a threshold $t$ is exceeded in the $z(x)$ field, we get the target which we will term "massive ore". The
massive ore is shown in Figure 1D and is the part of the orebody that would be considered for mining. In this example, this
results in a single economic parameter: volume. We do not consider concentration, grade, or other economic parameters in
this paper, though the methodology does not prevent including them.

The next element is the set of measurements that are available to be taken. Measurements are indirect indicators of what is
desired: the economic parameters of the orebody, which in our setting is the orebody volume. Measurements generally do
not directly observe this value; however, they may reduce the uncertainty on it. Such uncertainty quantification is generally
conducted with Bayesian approaches. Bayesian methods require stating measurement likelihood functions and prior
distributions. In our setting, the various alternative realizations constitute samples of the prior. In this work, we consider
taking point measurements of the total variational field, as shown in Figure 1C. We also consider taking only one
measurement at a time because measuring may be expensive, and the results may inform where to best take the next
measurement. Note that in this work, we will not perform traditional geostatistical conditional simulation using the





measurements as hard data, because the function $m(x)$ is stochastic as well. Instead, we will solve Bayesian inverse problems that aims to infer $m(x)$ and $r(x)$ jointly from data. $z(x)$ represents the exhaustive set of observations that could be acquired. In the real world, measurements may have various degrees of noise (e.g. geophysical survey vs. borehole data). In

this work, we assume that the noise on the point measurements is negligible, but that only a small area is directly observed. Measurement noise can be integrated into the Bayesian inverse problem, but our paper does not focus on it.

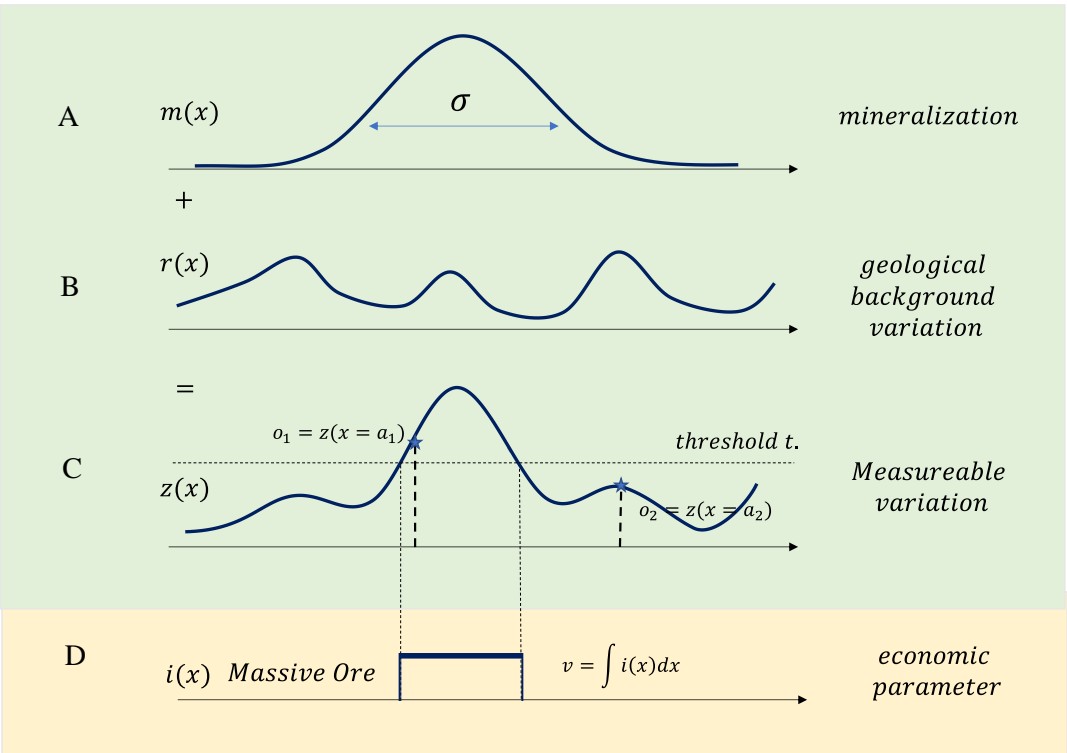

**Figure 1. Example 1D Mineralization. Sub-figure (A) shows a mineralization that is altered by geological background variation**
**(B), resulting in the measurable variation (C). The massive orebody (D), whose volume is the economic parameter of interest, exists**
**at locations where z(x) exceeds a threshold value.**

We test the presented methodology on a 2D case that is analogous to the 1D example. The 2D case set-up is shown in Figure 2 and Figure 3. We define the mineralization $m(x)$ using a single uncertain parameter $\sigma$ that determines the width. We

assume $\sigma$ has a uniform distribution with known bounds. Geological variation is modeled using a Gaussian process with known mean and variogram. We generate the measurable fields $z(x)$ by adding various realizations of $m(x)$ to realizations of $r(x)$, as shown in Figure 2. Then after defining a threshold $t$, we obtain the massive ore field $i(x)$ with the volume $v$, as shown in Figure 3.



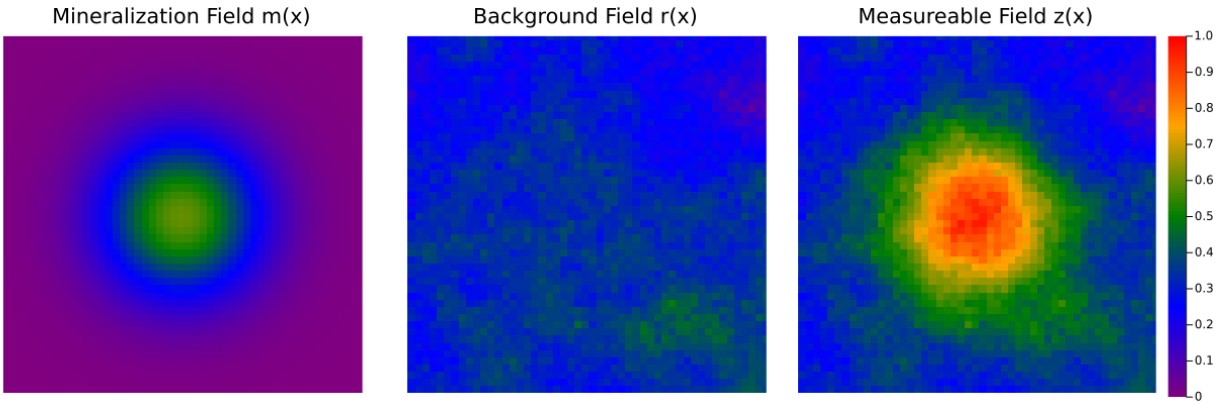


**Figure 2. Two-dimensional exploration problem. The mineralization field $m(x)$ (left), the background field $r(x)$ (center) are summed to create the measurable field $z(x)$ (right).**

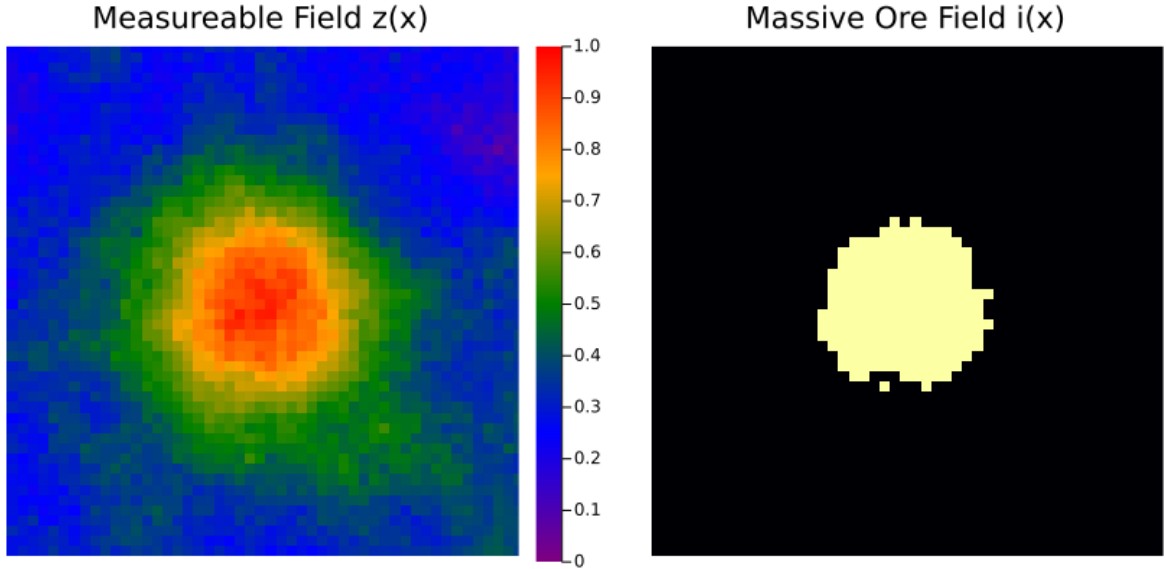

**Figure 3. Two-dimensional economic field. The massive ore field $i(x)$ shows where the measurable field $z(x)$ exceeds the economic threshold $z_{threshold}$.**

The question we will address is: what is the optimal sequence of data acquisition that best informs "mine" vs "do not mine"

decision?






## 3 Notational aspects

In this paper, we will need to merge nomenclature and mathematical notations of two different domains: geosciences/geostatistics and artificial intelligence (AI). Here we list some nomenclature from each field that describe the same concept (see also Table 1).

- A state = an instantiation of a set of parameters describing the world. For example, a geostatistical realization is a set of geological parameters representing the "state" of the subsurface in a gridded model. A state is referred to as $s$.

- Belief over a state = probability distribution of instantiations of a set of parameters. In probability theory, one defines over all possible outcomes of a geological model a probability density. This density is very high-dimensional in our setting. In AI ones uses $b(s)$, while in probability parlor, this is referred to as $f(s)$.

- Belief update = Bayesian update. A belief update requires stating the prior and the likelihood model. The likelihood in AI is termed the observation model $O(o|s, a)$, while in Bayesian terminology one uses $f(o|s)$. Note that in AI an additional "conditioning" is added as $a$, which represents the action by an AI agent. This is accounts for the fact that actions are taken in sequences. $O(o_{t+1} | s_{t+1}, a_t)$ is the likelihood of the observation at measurement $t + 1$, given the state at $t + 1$ and action at $t$.

- Observation space: the set of all possible outcomes of the measurements. In AI observations are denoted as $o$, while in Bayesian nomenclature these are termed data $d$.

| AI Terminology | Geosciences Terminology | Definition |
|---|---|---|
| State: s | Realization: $z(x)$ | The (possibly unknown) subsurface geological parameters |
| Action: $a$ | Take measurement | Measure $z(x)$ at $x$ |
| Observation: $o$ | Measurement | Measured value of $z(x)$ |
| Belief: $b(s)$ | Probability density over $z(x)$ | A probability distribution over the possible geological parameter realizations |
| Belief Update | Bayesian Posterior | Updating the distribution over geological parameters given new information according to Bayes' rule |

Table 1: comparison between AI and geostatistical nomenclature





## 4 Methodology

### 4.1 Partially observable Markov decision processes

This work frames mineral exploration as a sequential decision process. In a sequential problem, a decision-making agent must take a sequence of actions to reach a goal. Information gained from each action in the sequence can inform the choice of subsequent actions. An optimal action sequence will account for the expected information gain from each action and its impact on future decisions. This type of conditional planning may be referred to as closed-loop or feedback control. We will use the mineral-exploration problem outlined above as a working example for the remainder of this section.


   A sequential decision problem can be modeled formally as a Markov decision process (MDP). An MDP is a mathematical description of a sequential decision process defined by a collection of probability distributions, spaces, and functions. The full MDP is typically defined by the tuple $(S, A, T, r, \gamma)$. The state space $S$ is the space of all states that the decision-making problem may take at any step. In the mineral exploration process, the state is defined by the geological

model of the subsurface deposit as well as the locations of measurements. The action space $A$ defines the set of all actions that the agent may take. In the mineral exploration problem, this would be the set of all locations that the agent may acquire measurements (data). The transition model $T(s_{t+1} \mid s_t, a_t)$, is the probability distribution over the next time step state $s_{t+1}$, conditioned on the current state and action. The step $t$ refers to the sequential actions and belief updates. The MDP formulation assumes that the state transition is fully informed by the immediately preceding state and action, which is the

Markovian assumption. The transition model may be deterministic.

   The reward function $r(s_t, a_t, s_{t+1}): S \times A \times S \rightarrow R$ gives a measure of how taking an action from a state contributes to the utility of the total action sequence which the agent seeks to maximize. The objective of an agent in an MDP is to maximize the sum of all rewards accumulated over an action sequence. To preference rewards earlier in the process, a time discount factor $\gamma \in (0,1]$ is used. The goal of solving an MDP is to maximize the sum of discounted rewards accumulated

from a given state, defined as

$$\sum_{t=1}^{T} \gamma^{t-1} r(s_t, a_t, s_{t+1})$$

for a decision process with $T$ steps. The sum of discounted rewards expected from a state is defined as the *value* of the state $V(s)$. Given that the exact state transitions are not generally known in advance, the optimization target of solving an MDP is to maximize the expected value.


   In many decision-making problems, such as all subsurface problems, the state at each time step (the geological model) is not fully known. In this case, agents make decisions based on imperfect observations of the relevant states of their environments. Sequential problems with state uncertainty are modeled as *partially observable* Markov decision processes





(POMDPs). POMDPs are defined by the MDP tuple plus an observation space $O$ and an observation model $Z(o_{t+1} \mid s_{t+1}, a_t)$. The observation space defines all the observations that the agent may make after taking an action. Observations are generally noisy measurements of a subset of the state. The observation model defines the conditional distribution of the observation given the state and action. In the mineral exploration problem, an observation would be the mineral content of the core sample taken at that time step.

To solve a POMDP, an agent must account for all the information gained from the sequence of previous observations when taking an action. It is common to represent the information gained from an observation sequence as a *belief*. A belief is a probability distribution over the unknown state of the world at a given time step. At the beginning of the decision-making process, the agent will start with a belief that is defined by all *prior* knowledge of the state available before making any observations. With each observation made, the belief is updated, typically using a Bayesian update as

$$b'(s_{t+1}) \propto L(o_{t+1} \mid s_{t+1}, a_t) b(s_{t+1}).$$

A belief may be an analytically defined probability distribution or an approximate distribution, such as a state ensemble updated with a particle filter.

Each decision in the sequence is made using the belief updated from the preceding observation. The process is depicted in Figure 4. An optimal choice in a sequential problem should consider all subsequent steps in the sequence. However, the number of trajectories of actions and observations reachable from a given state grows exponentially with the length of the sequence. As a result, optimizing conditional plans exactly is generally intractable. Instead, most POMDPs are solved approximately using stochastic planning and learning methods.





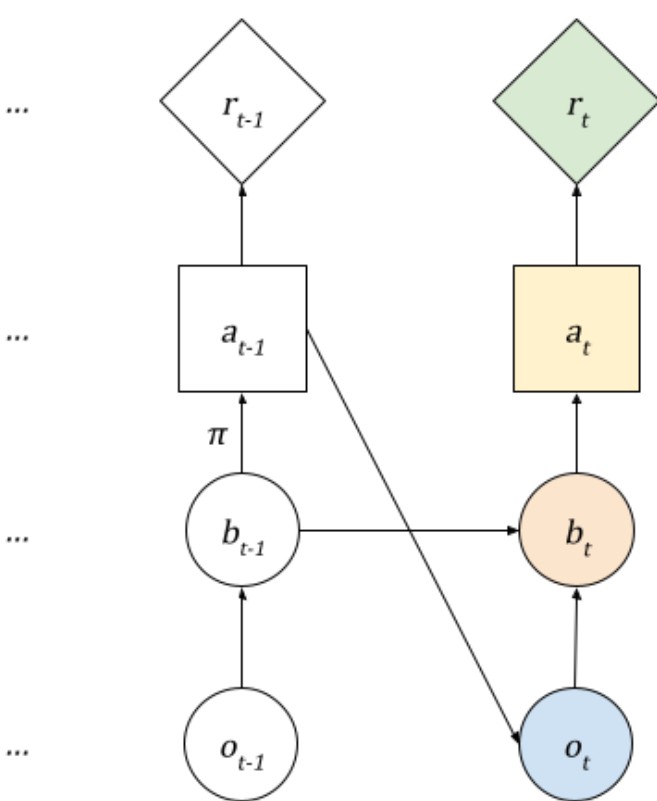

**235**   **Figure 4: Exploration Markov Decision Process. Each decision step, the agent selects an action $a_t$ based on its current belief over the world state using a planner ($\pi$). The agent takes the action in the world and observes some new data $o_{t+1}$. This data is used to update the belief $b_{t+1}$ for the next step. Each action results in a reward.**

Monte Carlo tree search (MCTS) is a class of stochastic planning algorithms that is commonly used to solve MDPs

and POMDPs. MCTS methods solve for actions each time a decision is made by simulating the potential outcome of available action sequences. It uses the simulations to estimate the expected value of each available action and then recommends the action with the highest expected value. Each simulated trajectory is recorded in a tree graph, as shown in Figure 5.. Each time a simulation is generated, the trajectory is added to the tree. Future action sequence trials are guided by the information in the tree at the start of that trial. MCTS algorithms are considered *online* planners, since they solve for an

optimal action from a given starting state, and therefore require computation every time a decision is made.



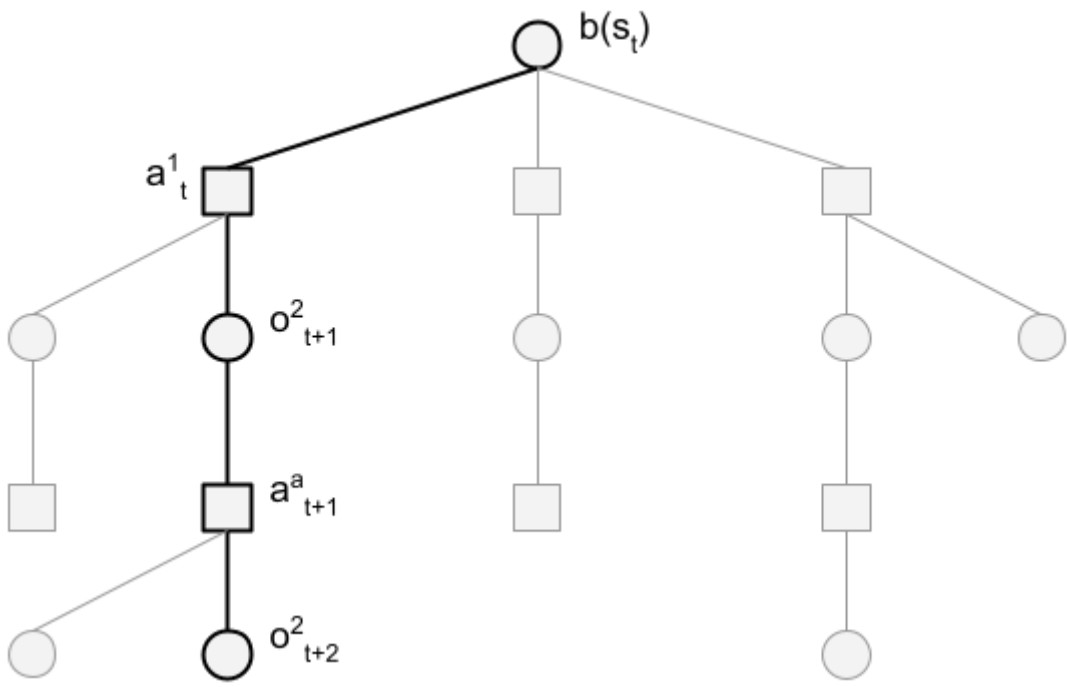

**Figure 5: Monte Carlo search tree. Each simulation in an MCTS algorithm is encoded into a search tree. The example tree is rooted at the belief, $b(s_t)$ given at the start of search. Paths from the root to a leaf of the tree represent a simulated trajectory of alternating actions, $a^i{}_t$ and observations, $o^i{}_t$. An example trajectory in the tree is shown in bold.**


### 4.2 A POMDP for resource exploration

We propose formulating the mineral exploration problem as a sequential decision problem. A sequential plan allows information from each measurement in the sequence to inform the choice of subsequent measurements.

We now return the template example introduced in Figure 1 and state the elements of the POMDP.

*State Space* ($S$): The state is a combination of a realization of the unknown subsurface geology (a geostatistical model) and any other environment factors that may constrain or affect the outcome of the measurements to be taken and the rewards gained.

• Example POMDP: The state space is the combination of the sub-surface state space and the measurement state space. The subsurface state in the case of Figure 2 is the combination of $m(x)$ and $r(x)$. The measurement state defines the location of all previously acquired measurements.





*Action Space* ($A$): The action space defines the set of measurement actions that can be taken at every step. The action space should also include MINE, and ABANDON (do not mine) actions. These actions allow the agent to terminate the measurement campaign.

- Example POMDP: The action space is the set of all locations at which a measurement may be acquired in the exploration area, along with the MINE and ABANDON actions. Each measurement action is defined by the target measurement location. Taking an action $a$ signifies measuring $z(x)$ at $x = a$. Available measurement locations are defined on a regular cartesian grid, and two measurements may not be drilled closer than some minimum distance from one another. The minimum distance may be set to zero to represent an unconstrained set.

*Observation Space* ($O$): The set of measurements values that may be observed from an action. The observation space may be composed of heterogeneous observation types to account for different measurements that may be taken; for example, to account for geochemical surface data and drill-core sample data.

- Example POMDP: The mineralization $z(x)$ measured at a targeted location is defined as a scalar value.

*Observation Model* ($L$): the observation model defines the effect of sensor and other noise on the data generated by measurements. In the case that observations can be treated as noiseless, the conditional distribution can be defined by the Dirac as $L(o \mid s', a) = \delta(o - g(s', a))$, where $g(s', a)$ is a deterministic function mapping the state and action to the observation. In Bayesian literature $g$ is also termed the data forward model.

*Transition Model* ($T$): The transition model defines how the state evolves as a result of actions. In our setting, the subsurface state does not change because of measuring actions, and only measurement state elements will be updated. The transition model can also be used to constrain the actions that are available at each step, by setting the transition probabilities to 0 for disallowed actions.

- Example POMDP: The measurement state is updated with newly selected action locations. Later, we will test two different transition models. One model does not constrain the available actions and a second constrains the action space to measurement locations that are no further than a distance $\delta$ away from the previous measurement. The purpose of doing so is to illustrate that the methodology allows for action constraints.

*Reward Function* ($r$): The reward function defines a cost for each measurement action taken and a reward for the final MINE or ABANDON decision. The reward function takes the following form

$r(s, a) = Cost(s, a) \; if \; a \in A_{Measurements}$

$r(s, a) = 0 \; if \; a = ABANDON$

$r(s, a) = Profit(s) \; if \; a = MINE$





where $Cost(s, a)$ defines the cost of taking a measurement, $Profit(s)$ defines the profit from mining a deposit, and $A_{Measurements}$ is the set of measurement actions.

- Example POMDP: Each measurement has a fixed cost, and the profit is a simple function of the amount of ore

present $v(s)$ (Figure 1D) and a fixed extraction cost, as shown below.

$$Cost(s, a) = c_{Measurement}$$

$$Profit(s) = v(s) - c_{Extraction}$$

Discount Rate ($\gamma$): The discount rate defines a time discount rate for the costs and and profits and is used to calculate the net present value (NPV) of the measurement campaign.

- Example POMDP: We use a discount rate of 0.99

### 4.3 Solving the POMDP

In this section, we present a method to solve the example 2D mineral exploration POMDP. The methods presented may be generalized to additional mineral exploration problems. Algorithms to solve POMDPs can typically be applied to any valid POMDP model, though with differing effectiveness. The remaining subsections are divided into the tasks required to solve the POMDP: belief updating and searching over the large, combinatorial space of possible action sequences.

### 4.3.1 Belief Modeling and Updating

Belief updating in AI is the equivalent of inverse modeling in the geosciences. In our setting, we have indirect measurements $o(z(x))$ of the state variables $m(x)$ and $r(x)$. We have assumed that $m(x)$ can be modeled with a single parameter, σ that is distributed uniformly over a known range. We also assume that $r(x)$ can be modeled as a Gaussian process with known mean $\mu_r$ and covariance $C_r$. The subscript $t$ denotes the step (iteration) of the decision-making process. After step $t$, a total of $t$ measurements have been taken and we denote the set of all measurements taken up to that point as

$$o_{1:t} = \{o(x_\alpha), x_\alpha = 1, \dots t\}$$

The observed measurements are dependent upon both random functions, $m(x)$ and $r(x)$, hence a traditional conditional simulation cannot be directly applied. Instead, we formulate this problem as a hierarchical Bayes' problem by factoring the joint distribution into

$$f(m(x), r(x)|o_{1:t}) = f(m(x)|o_{1:t}) \times f(r(x)|m(x), o_{1:t})$$



Samples are generated from this distribution hierarchically by first drawing a sample from the distribution over $m(x)$ and then using the resulting sample to draw from the conditional distribution over $r(x)$. We model the belief $f(m(x)|o_{1:t})$ as a particle set and update it using an importance resampling particle filter (Del Moral 1996, Liu et al. 1997). The conditional belief $f(r(x)|m(x), o_{1:t})$ is modeled as a conditional Gaussian process.

   A particle set is an ensemble of realizations of the state variable with a sample distribution approximating the true
state distribution. The initial particle set is generated by first sampling an ensemble from the uniform prior distribution. For an $n$ particle set, this corresponds to an ensemble of $\left(m^i(x), r^i(x)\right), i = 1, \dots n$ where each particle is equiprobable.

   When new information $o_t$ is observed, the particle filter updates the belief by updating the ensemble such that the new particles are sampled according to the posterior distribution $f(m(x)|o_{1:t})$. To do this, a posterior weight is calculated for each particle according to Bayes' rule as

$$w^i \propto f(o_t|m(x), o_{1:t-1})$$

Note that each particle is treated as equiprobable in the particle set, so the prior probability is dropped in the above expression. The observed measurement $o_t$ is determined by the sum of $m(x)$ and $r(x)$ at the location of the measurement. We denote these values as $o_t^m$ and $o_t^r$, respectively, such that $o_t = o_t^m + o_t^r$. Using this notation, we can decompose the particle weight function into

$$w^i \propto f\left(o_m^t|m(x)\right) \times f(o_r^t|m(x), o_{1:t-1})$$

Because the value of $o_m^t$ is completely determined by $m(x)$, we can simplify this further to

$$w^i \propto f(o_t - o_m^t|o_{1:t-1} - m(x))$$

which is given by the Gaussian process model conditioned on the difference between the previous measurements and the $m(x)$ values at their corresponding locations.

Once a weight has been calculated for each particle in the set, a new ensemble is generated. The new set is generated by sampling $n$ particles from the weighted set, with each particle being sampled with probability given by its weight. For each particle sampled, a new $r(x)$ field is generated with conditional Gaussian simulation, conditioning on the residual of the observed measurements and the sampled $m(x)$ field as

$$r(x) \sim N(\mu_r, C_r | o_{1:t} - m(x))$$

Sampling a particle ensemble with replacement in this way can lead to degeneracy, in which only a few values of $m(x)$ are represented in the filtered ensemble. To prevent this, particles that are duplicated in the ensemble are perturbed slightly by adding zero-mean Gaussian noise to the $\sigma$ parameter generating $m(x)$. The complete belief update is summarized in pseudocode in *Algorithm 1* (Table 2) and described in text below.






---

**Algorithm 1** UPDATEBELIEF

---

```
function UPDATEBELIEF(b, a, o)

  O ← b_o U{o}
  A ← b_a U {a}
  W ← ()
  for p_i in b                            Calculate Particle Weights

    r_i ← o - m_i(x_a)
    w_i ← N(r_i; μ_i(x_a), σ_i(x_a))
    APPEND w_i to W

  η ← 1/sum_i w_i
  for w_i in W                            Normalize Weights

    w_i ← η w_i

  D ← {}
  P ← {}
  while |P| < |b|                         Resample Particles

    p ← SAMPLE(b, W)
    if d in D

      d ← d + e ~ N(0, σ²_n)

    m(x) ← f(x; d) forall x
    R ← O(x_a) - m(x_a) forall x_a in A
    r(x) ~ GP(A, R)                       Conditional Gaussian process
    z(x) ← m(x) + r(x)
    p' ← (d, z(x))
    P ← P U {p'}

  b' ← (P, O, A)
  return b'
```

---

**Table 2: pseudo algorithm for model inversion (belief update) using a hierarchical particle filter**

### 4.3.2 Online Monte Carlo Planning

To solve the POMDP, we search for the optimal action at each step using a variant of POMCPOW (Partially

Observable Monte Carlo Planning with Observation Widening; Sunberg and Kochenderfer, 2018), a Monte Carlo tree search algorithm for POMDPs. Each time step $t$, the POMCPOW algorithm builds a tree of possible trajectories, with the root node of the tree representing the belief $b_t$. The full tree is constructed before taking any action at that step. The action with the highest estimated value is then returned from the search process.





POMCPOW generates a fixed number of trial trajectories $m$, by sampling $m$ states from the root belief. For each
sampled state, POMCPOW simulates taking a series of actions $a_t, \dots, a_{t+k}$, and encodes the resulting series of observations
as a branch of the tree. For each action visited along the branch, POMCPOW updates the estimate of the expected value of
taking that action in the sequence using the rewards simulated in that trial. We modified the baseline POMCPOW algorithm
by replacing the Monte Carlo value estimation with generalized mean estimation. The value of an action node in a tree is
then given as

$$\bar{Q}(b, a) = \frac{1}{n} \sum_{b' \in Ch} \bar{V}(b')$$


where $Ch$ is the set of $n$ child belief nodes of action node, $a$. The $\bar{V}(b)$ term gives the estimated value of each belief node,
defined as

$$\bar{V}(b) = \left( \frac{1}{n} \sum_{a \in Ch} \bar{Q}(b, a)^\alpha \right)^{1/\alpha}$$

where $Ch$ is the set of child action nodes of the estimated belief node. The value $\alpha > 0$ is a parameter, where values of $\alpha >$
1 more heavily weight actions with higher estimated values. We used $\alpha = \infty$, which resulted in backing up the maximum
action node estimate at each belief node.

Each step of a simulated trial, POMCPOW simulates taking the action with the highest upper confidence bound on
its estimated value. In this way, POMCPOW optimistically explores the action space. This strategy has been proven to
converge to the optimal action in the limit of infinite samples. After all $m$ trials have been generated, POMCPOW returns
the root node child action with the highest estimated value.

For POMDPs with large action spaces, POMCPOW limits how often new actions can be added to the search tree
through a progressive widening rule. Under progressive widening, the total number of child action nodes that a given belief
node may have, is defined as a function of the total number of times that node has been visited in previous trials. The limit is
defined as $C_{max} = kn^\alpha$, where $n$ is the total number of previous visits. Actions added to the tree are sampled according to a
stochastic policy. We defined the k-σ upper confidence bound for each point in the exploration area as $UCB(x) = m(x) +$
$\mu(x) + k\sigma(x)$, where $\mu$ and $\sigma$ are given by the distribution of the parent node belief. Actions were then sampled in
proportion to the UCB value at the target location. Intuitively, this guided POMCPOW to search actions that had both high
expected value, and high uncertainty.

### 4.4 Illustration Case


In this section, we present the result of solving the problem for the mineral field shown in Figure 6, below. In all
problems, rewards are measured in units of massive ore, where one pixel in the massive ore map (Figure 3) represents one

unit of ore. In all the problems studied, the massive ore threshold was set to 0.7 and the extraction cost was set to 150 units. This example case has a total volume of 158 units massive ore, making it a marginally profitable case. The measurement cost
was 0.1 units per measurement taken. In this example, we constrained the measurements to be taken a maximum distance of 10 distance units away from the previous measurement, where each pixel is one distance unit.

Figure 7 shows the mean and standard deviation mineralization $z(x)$ at each point in the exploration area calculated from the initial belief ensemble before any measurements have been taken. The histogram in Figure 8 shows the distribution of massive ore quantities for the realizations in the ensemble. The vertical line shows the 158 massive ore volume of the
illustration case realization.

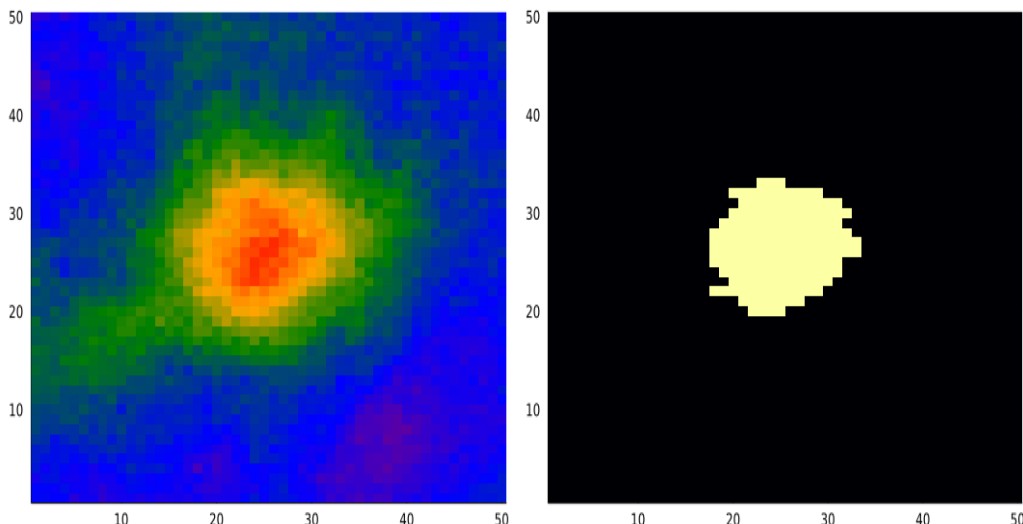

**Figure 6: Illustration case. The left figure shows the mineralization $z(x)$ of the example case. The right figure shows the massive ore mass of the mineral field $i(x)$.**




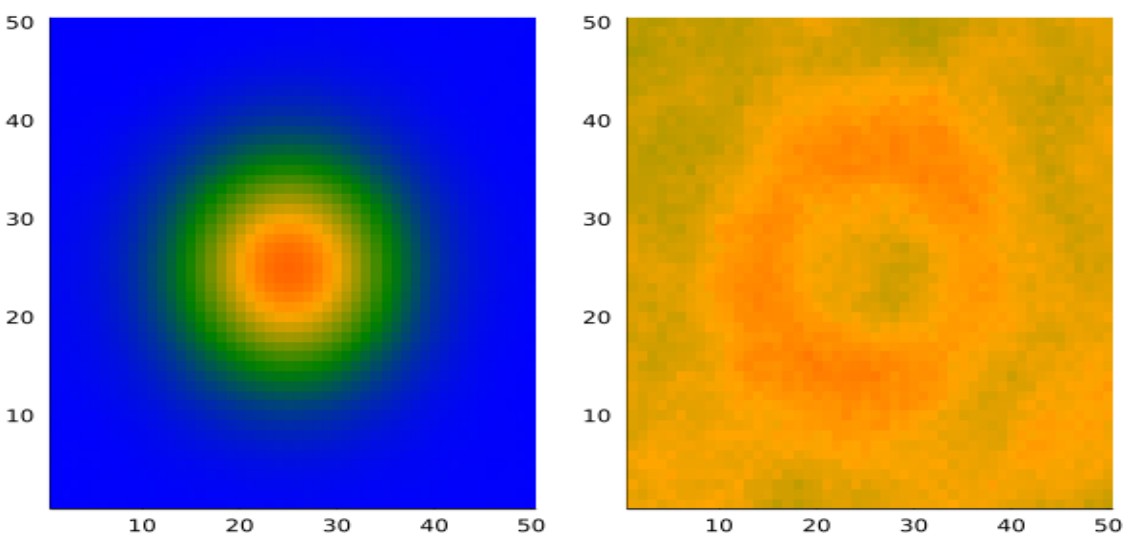

**Figure 7: Initial ore belief. The left figure shows the mean mineralization from the prior belief at each point in the exploration area. The figure on the right shows the marginal standard deviation of mineralization at each point.**

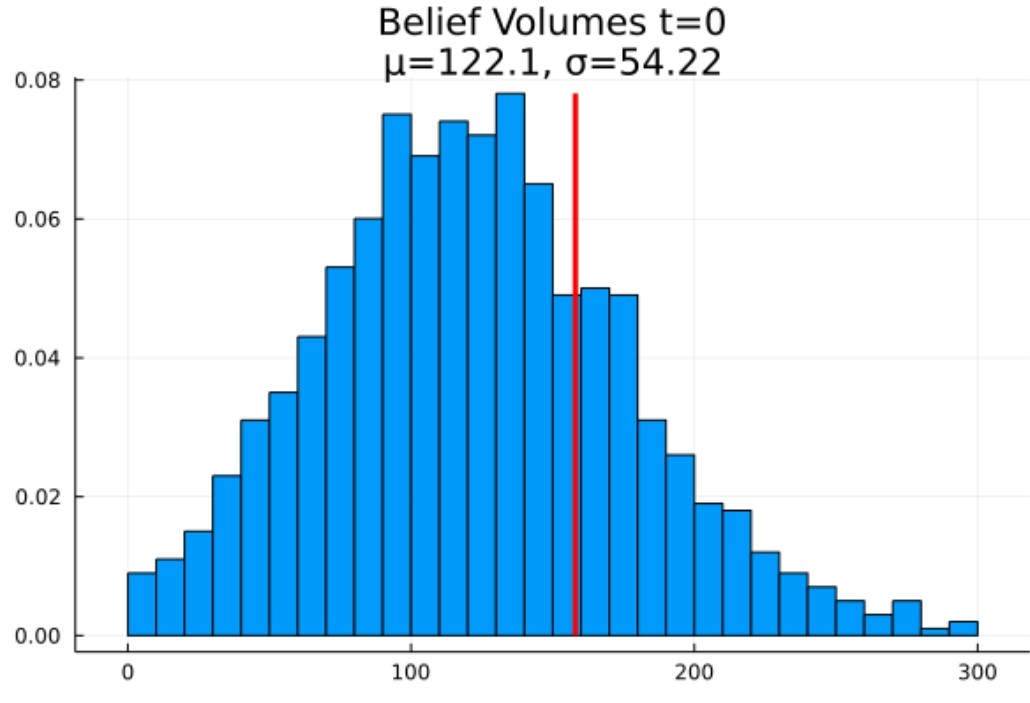


**Figure 8: Initial belief ore histogram. The figure shows the distribution of massive ore volumes in the initial belief ensemble. The vertical line shows the actual volume of ore in the illustration case.**



We ran POMCPOW for 10,000 simulations per-step. The resulting actions taken in the first five steps are shown in Figure 9, below. As can be seen, the deviation of the belief over the ore quantities decreases as measurements are taken, and the expected value tends toward the true value. The agent tends to take an "extent finding" approach, where it alternates taking actions closer and then farther from the expected center of the orebody. This pattern may be interpreted as searching for the maximum extent of the ore-body edge.





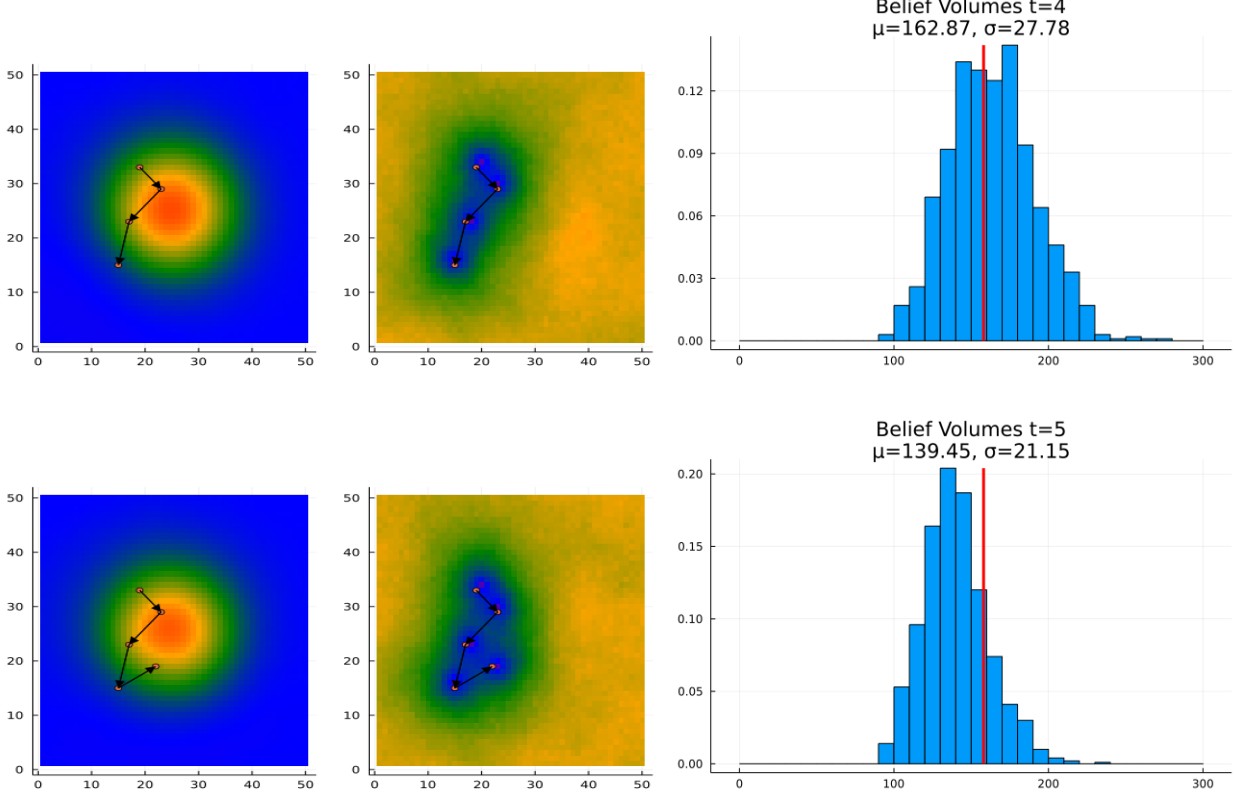

**Figure 9: Initial measurement trajectory. Each figure shows the belief resulting from the measurements taken by the agent. The circles show the locations at which measurements were taken. The arrows indicate the sequence in which actions were taken.**

The complete 22 measurement trajectory is shown in Figure 10 below along with the final histogram. At the conclusion of the measurements, the algorithm correctly decided to mine the deposit. As can be seen, at the time it made its decision, the expected value of the ore-quantity was approximately one standard deviation above the extraction cost threshold of 150. The agent did not stop exploring once the expected value exceeded the threshold, but only once it had exceeded by a significant threshold. This suggests that the agent would stop only when the value of the information gained

by a measurement was exceeded by the cost of the measurement.





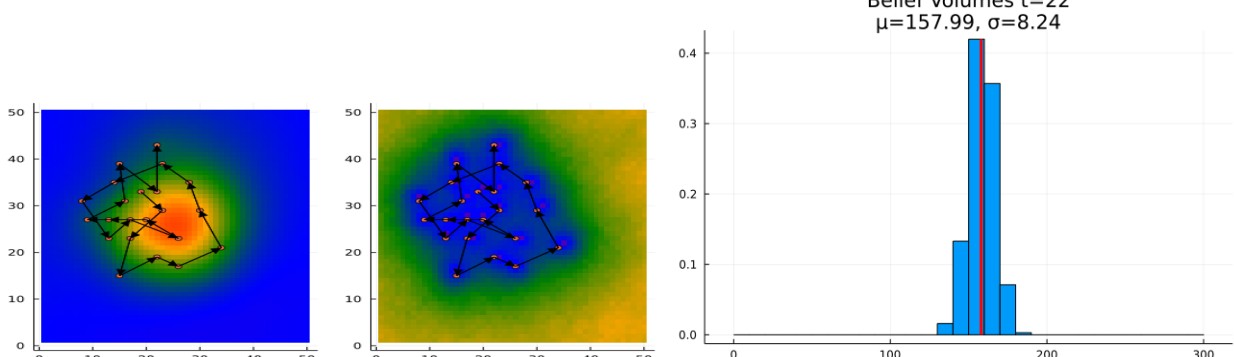

**Figure 10: Complete measurement trajectory. The figure on the left shows the complete trajectory of all measurements taken in the illustration case. The figure on the right shows the resulting histogram.**

## 5 Experiments and Comparison with Baseline Methods

### 5.1 Overview of Test Cases

To test the proposed approach, we conducted experiments on a variety of problem configurations. For these experiments, we tested three different ore-settings.

1. Single body, fixed position: A single mineralization process generated an ore body with a known centroid location at the center of the exploration domain.
2. Single body, variable-position: A single mineralization process generated an ore body with an unknown centroid location somewhere in the exploration domain.
3. Two body, variable-positions: Two mineralization processes generated orebodies, both with unknown centroid locations within the exploration domain.

The illustration case previously presented was from the single body, fixed-position problem configuration. Examples of the single body, variable-position and two body cases are shown in Figure 11. For each problem configuration we tested the POMCPOW agent with measurements constrained to a distance of 10 units from the previous location and without constraints on measurement location.



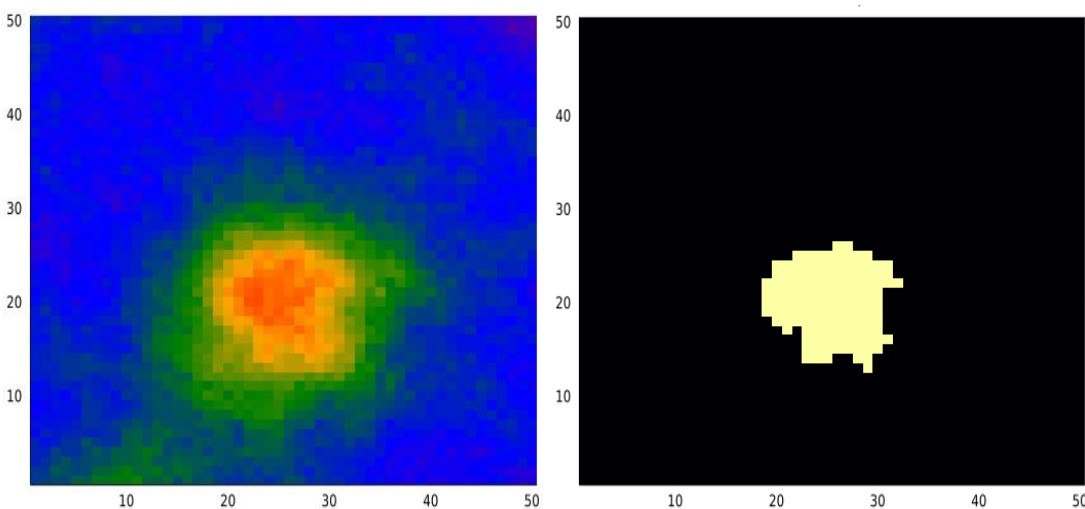


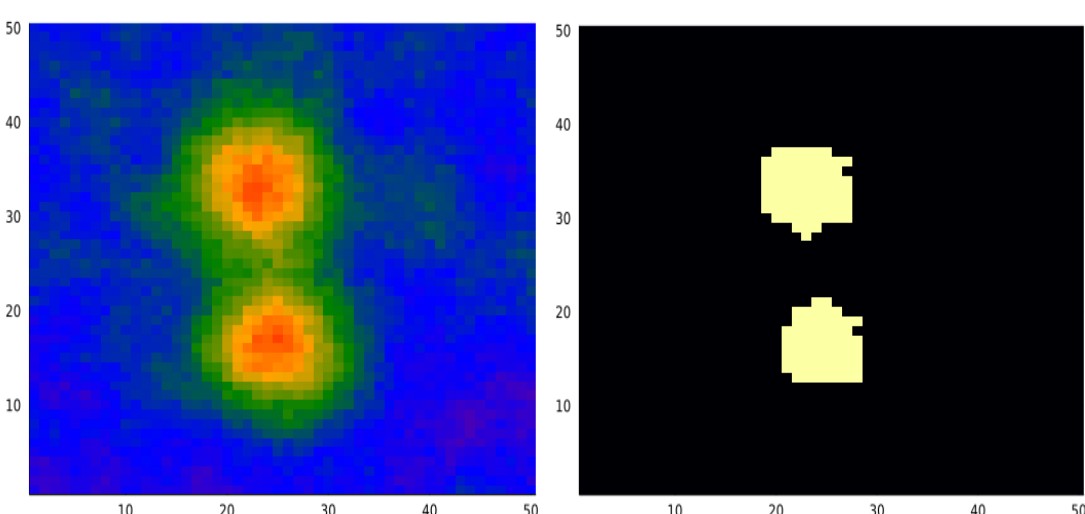

**Figure 11: (Top row): Single body, variable location realization. The left figure shows the mineral field generated by a primary process with a randomly selected centroid location. The right figure is the corresponding massive-ore map. (bottom row) Two body realization. The left figure shows the mineral field generated by two primary processes, each with a randomly selected centroid location. The right figure is the corresponding massive-ore map.**


We also tested the performance of POMCPOW against a baseline grid-pattern approach. In this method, measurements were taken at locations defined by $k$-by-$k$ grids, as shown in Figure 12. Each grid pattern covers a square area located at the center of the exploration domain, with measurement coordinates taken at regularly spaced intervals along the

cartesian directions of the grid. We solved for the optimal grid area for a 3-by-3 measurement grid by minimizing the





expected standard deviation of the resulting belief. We solved for this value by first optimizing with Nelder-Mead simplex search (Nelder 1965) on the continuous range [5, 50] and then rounding the resulting value. The grid area was set to 30-by-30 for all grid patterns.

We tested grids with 4, 9, and 16 measurements, as well as a single point fixed at the center of the exploration area.
We also tested a baseline in which measurement locations were selected at random at each step. This allows us to understand the improvement of the approaches relative to an achievable lower-bound.

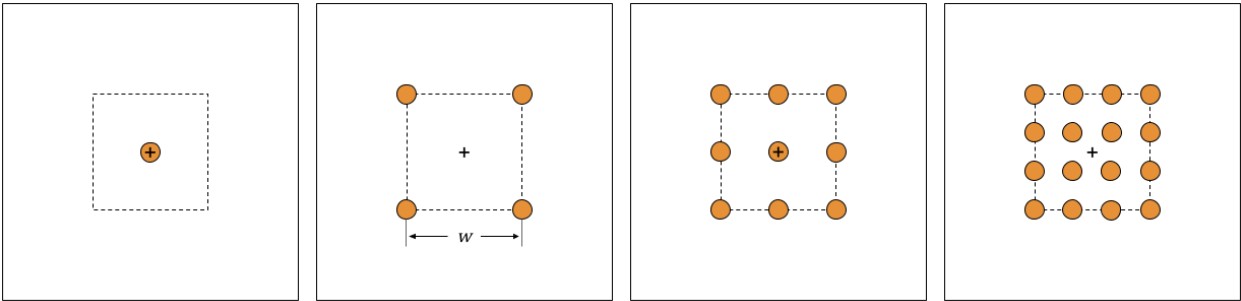

**Figure 12: Baseline grid patterns. The figures show the baseline grid patterns for 2-by-2, 3-by-3 and 4-by-4 grids, each with a total**
**of 4, 9, and 16 measurements respectively. The grids cover the extent of a *w*-by-*w* area in the center of the exploration domain. A single measurement at the center of the domain is also shown in the leftmost figure.**

We ran Monte Carlo tests on the problem configurations described. For each case, we generated a set of 100 mineral-field realizations, each one assumed as a possible truth. For each realization, measurements were taken according to
the constrained and unconstrained POMCPOW solvers, the grid policy, and the random policy. The change in mean error and standard deviation for all the approaches was calculated. For the POMCPOW solver, we also measured the expected number of measurements as a function of the total deposit size, and the accuracy of the final MINE or ABANDON decision.

The data from the tests suggested that different behavior emerged through POMCPOW for cases that were non-economic, highly economic, and borderline economic. To investigate this, we solved one of each economic level for the
three deposit settings using POMCPOW with action constraints. At the end of this section, we present the results of these trials and a plot of the observed trend in the Monte Carlo data.

**5.2 Single Body, Fixed Location**

In this section, we present the results for the Monte Carlo tests on the case with a single, unimodal mineralization process located at the center of the exploration domain. For every solver, we measured the belief accuracy by calculating the
relative mean absolute error (RMAE) of the estimated deposit volume resulting from each measurement. The relative MAE is the estimate error relative to the true deposit volume and is defined as





$$RMAE = \frac{1}{n}\sum_{i=1}^{n}\frac{|\bar{v}_i - v_i|}{v_i}$$

where $\bar{v}_i$ and $v_i$ are the estimated and true deposit volumes for trial $i$, respectively. We calculated the RMAE after each measurement was taken by the POMCPOW policies and the random baseline. We also calculated the RMAE after all

measurements were taken for the grid patterns with one, four, nine, and sixteen measurements. The resulting trends are shown in Figure 13 with one standard error bounds.

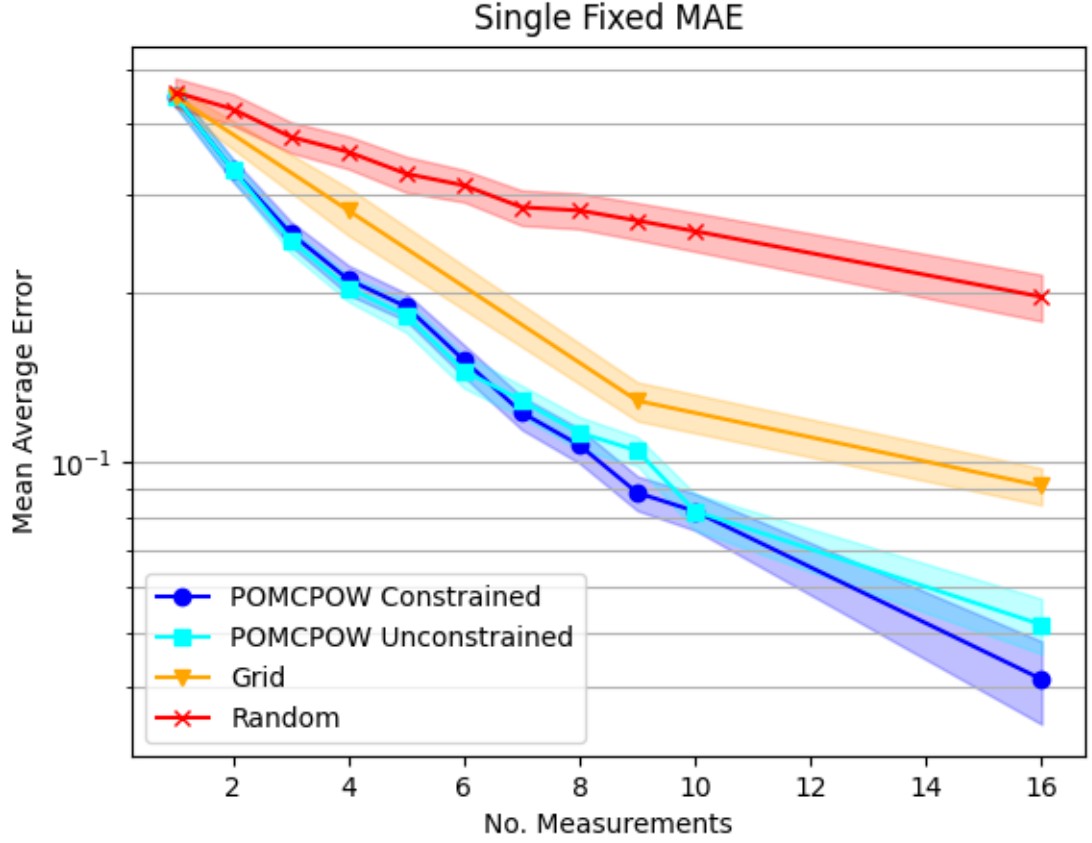

**Figure 13: Relative MAE single mineralization, fixed location. The plot shows the mean relative absolute error after a given number of measurements taken under each tested method. The mean absolute error is shown along with one standard error**
**bounds for each trend.**

We also measured the change in uncertainty (belief) by calculating the standard deviation resulting from each measurement. After each measurement, we calculated the ratio of the resulting volume standard deviation relative to the initial belief standard deviation (the Bayesian prior of volume). After measurement $t$ in the sequence, the standard deviation



ratio is given by $\frac{\sigma_t}{\sigma_0}$, where $\sigma_t$ is the belief standard deviation after the measurement (posterior standard deviation of volume),

and $\sigma_0$ is the standard deviation of the initial belief. We calculated this ratio after each measurement was taken by the

POMCPOW policies and the random baseline. We also calculated the ratio after all measurements were taken for the grid

patterns with one, four, nine, and sixteen measurements. The mean standard deviation ratios over the Monte Carlo trials for

each of the solvers is shown in Figure 14 along with one standard error bounds.

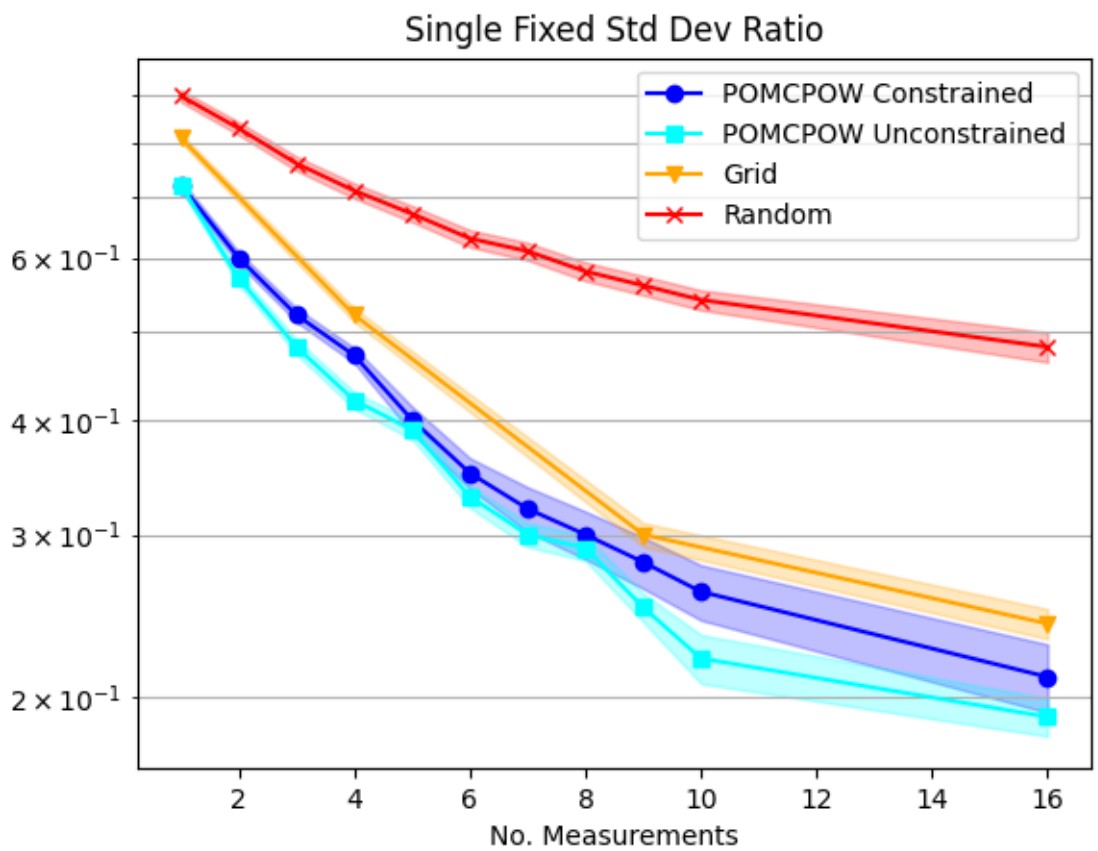


**Figure 14: Single Body, fixed location standard deviation ratios. The plot shows the mean standard deviation ratio after a given number of measurements taken under each tested method. The mean ratio is shown along with one standard error bounds for each trend.**

505        In addition to the belief trends shown above, we also further analyzed the behavior of the POMCPOW methods

with and without action distance constraints. For each, we examined the accuracy of the algorithm in making its final MINE

or ABANDON decision, as well as how many measurements it took before reaching a decision. We also looked at the

general trend in where it took measurements relative to the mineralization centroid location. These are presented in the

following sub-sections.




### 5.2.1 POMCPOW, Constrained Actions

The final decision results for the POMCPOW solver with constraints on the maximum distance between measurement locations is shown in Table 3, below. This table presents the proportions of profitable and unprofitable deposits that POMCPOW decided to MINE or ABANDON at the end of each trial. A deposit is profitable if the ore volume exceeds the extraction threshold. A decision to MINE a profitable deposit or to ABANDON an unprofitable deposit is considered correct. The total amount of ore in profitable deposits that was mined is also presented. The average number of measurements taken before making a decision is shown for each deposit type, and for all cases.

|  | Mined | Abandoned | Total | Accuracy |
|---|---|---|---|---|
| Profitable | **28** | 4 | 32 | 87.5% |
| Unprofitable | 2 | **66** | 68 | 97.1% |
| Total | 30 | 70 | 100 | 94.0% |
| Profitable Ore | **1097** | 57 | 1154 | 95.0% |
| Mean Measures | 7.8 | 5.9 | 6.5 | – |

**Table 3: Single Body, fixed location POMCPOW results with action constraints.**


Among the assumed "true" deposits, 32% are profitable. Among all the profitable cases, there is a total of 1154 units of ore, with POMCPOW deciding to mine 1097 units corresponding to 95% of profitable ore correctly extracted. On average, POMCPOW took 1.8 more measurements in profitable cases than in unprofitable cases.


POMCPOW was able to decide when to terminate taking measurements at any point during the campaign. If it did not decide to terminate, it was limited to a total of 25 measurements. Figure 15 below shows the histogram of the number of measurements before termination taken by POMCPOW over the Monte Carlo trials.





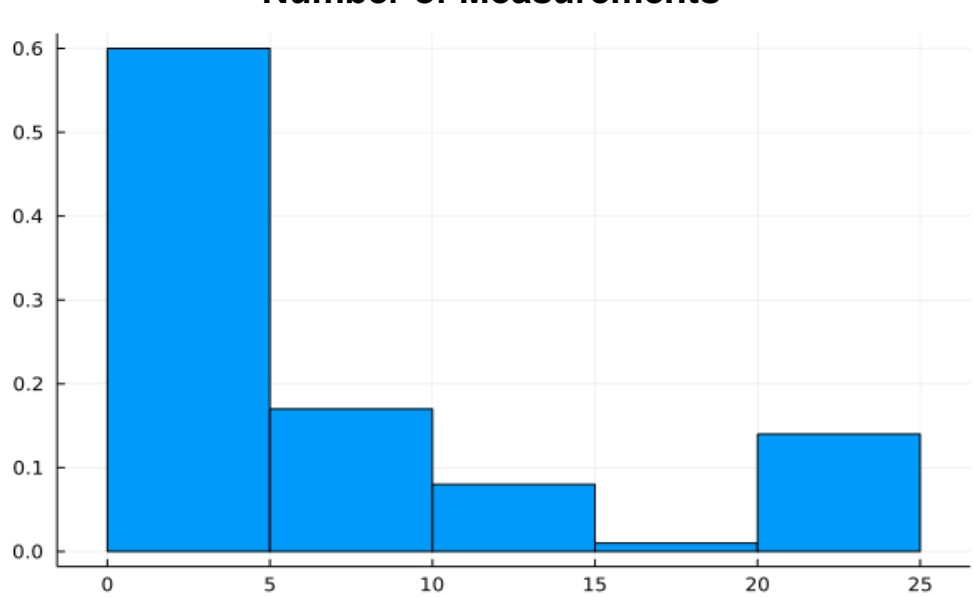

**Figure 15: Measurement histogram, POMCPOW with action constraints, single body with fixed location. This figure shows histogram of the number of measurements taken by the POMCPOW solver over all Monte Carlo trials. The trials were limited to 540 a maximum of 25 measurements.**

We recorded the distance between each measurement in the sequence and the center of the mineralization. The average distance for each point in the sequence is shown for ten measurements in Figure 16, along with one standard error bars. One notice how the agent starts away from the center of the orebody, then steps in toward the center, then gradually steps away 545 from the center.

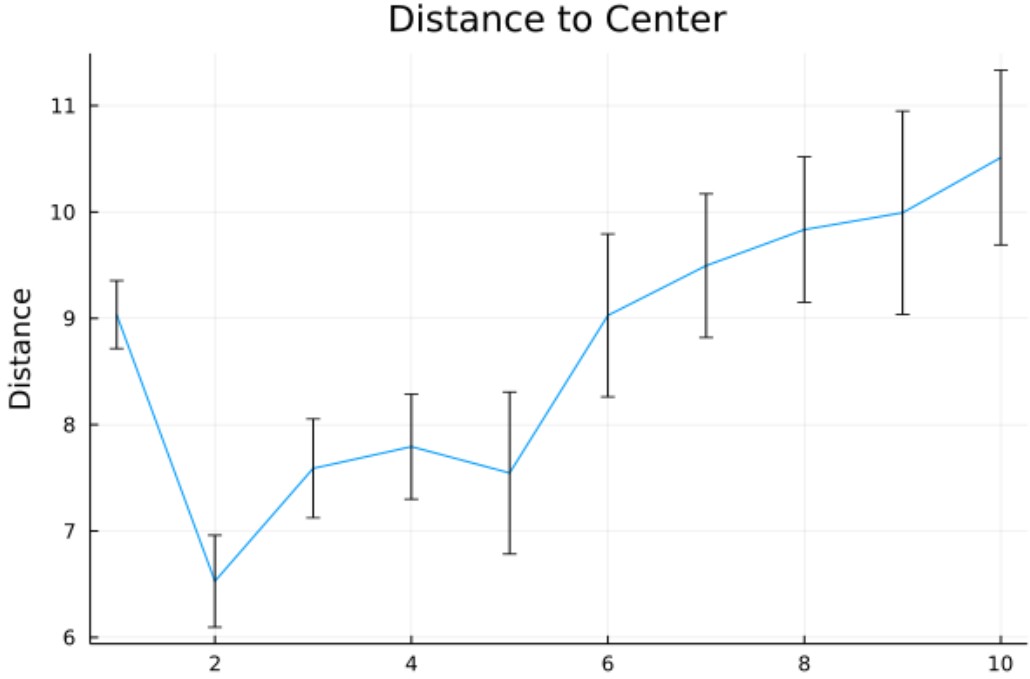

**Figure 16: Measurement distance to center, POMCPOW with action constraints, single body with fixed location. The plot shows the average distance between the measurement location and the mineralization center for the measurements at each time step. One standard error bars are also presented.**

### 5.2.2 POMCPOW, Unconstrained Actions

The final decision results for the POMCPOW solver with no constraints on measurement locations is shown in Table 4, below. The same set of trial deposits were used to test both the constrained and unconstrained cases. The same results as presented in the constrained case are presented here for the unconstrained case.

|  | Mined | Abandoned | Total | Accuracy |
|---|---|---|---|---|
| Profitable | **27** | 5 | 32 | 84.4% |
| Unprofitable | 5 | **63** | 68 | 92.6% |
| Total | 30 | 70 | 100 | 90.0% |
| Profitable Ore | **1058** | 96 | 1154 | 91.6% |
| Mean Measures | 7.6 | 5.9 | 6.4 | – |

**Table 4: Single Body, fixed location POMCPOW results without action constraints.**





Among all the profitable cases, there is a total of 1154 units of ore, with POMCPOW deciding to mine 1058 units
corresponding to 91.6% of profitable ore correctly extracted. On average, POMCPOW took 1.7 more measurements in
profitable cases than in unprofitable cases.

As in the constrained test, we plot the number of measurements taken before making the final decision in Figure 17,
below. We also present the average distance from the deposit center in Figure 18.


## Number of Measurements

**Figure 17: Measurement histogram, POMCPOW without action constraints, single body with fixed location. This figure shows histogram of the number of measurements taken by the POMCPOW solver over all Monte Carlo trials. The trials were limited to a maximum of 25 measurements.**




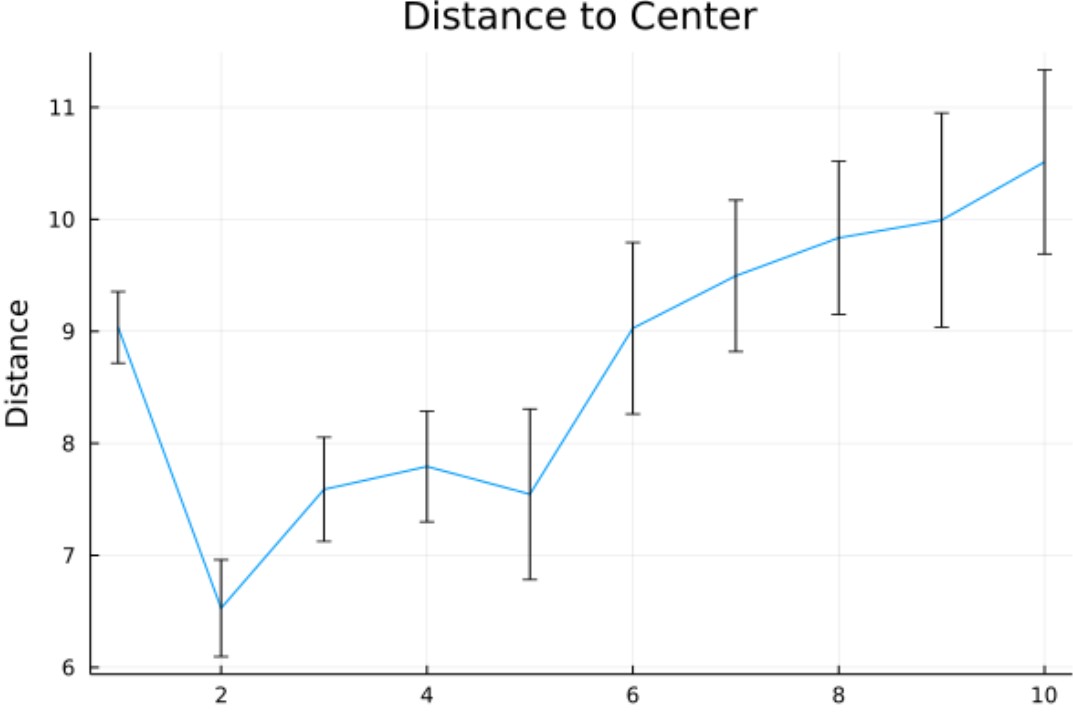

**Figure 18: Measurement distance to center, POMCPOW without action constraints, single body with fixed location. The plot shows the average distance between the measurement location and the mineralization center for the measurements at each time step. One standard error bars are also presented.**


**5.3 Single Body, Variable Location**

      In this section, we present the results for the Monte Carlo tests on the case with a single, unimodal mineralization process located at a variable, unknown point in the exploration domain. For every solver, we measured the belief accuracy by calculating the relative mean absolute error (RMAE) of the estimated deposit volume resulting from each measurement.

The resulting trends are shown in Figure 19 with one standard error bounds.

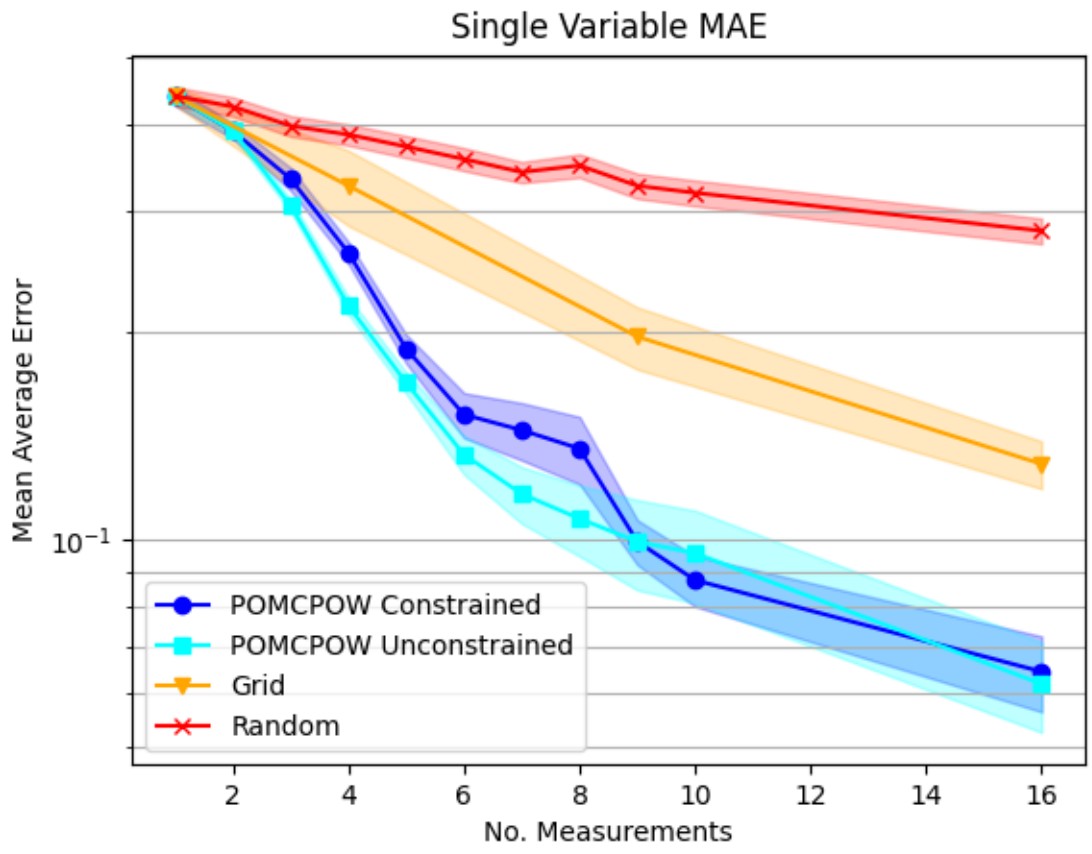

**Figure 19: Relative MAE single mineralization, variable location. The plot shows the mean relative absolute error after a given number of measurements taken under each tested method. The mean absolute error is shown along with one standard error**
**bounds for each trend.**

We also measured the change in belief uncertainty by calculating the standard deviation ratios of the belief volume estimate resulting from each measurement. The mean standard deviation ratios over the Monte Carlo trials for each of the solvers is shown in Figure 20 along with one standard error bounds.




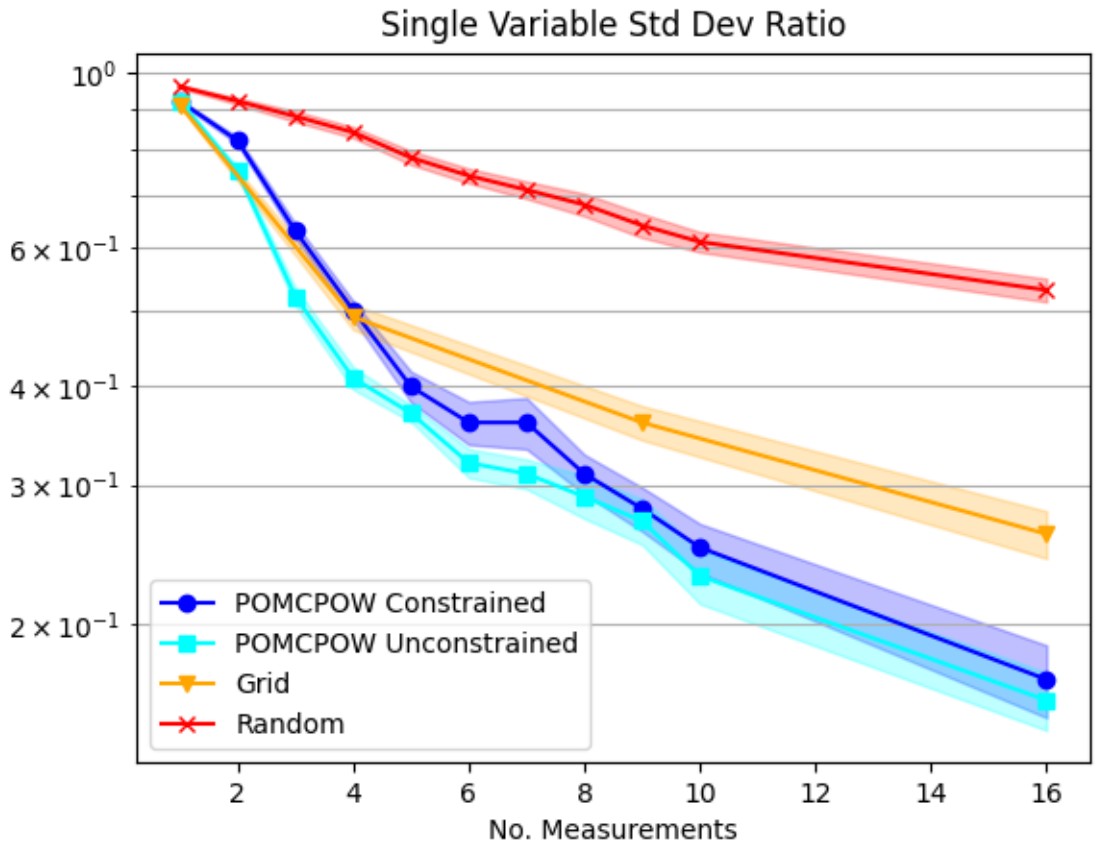

Figure 20: Single Body, variable location standard deviation ratios. The plot shows the mean standard deviation ratio after a given number of measurements taken under each tested method. The mean ratio is shown along with one standard error bounds for each trend.


### 5.3.1 POMCPOW, Constrained Actions

The final decision results for the POMCPOW solver with distance constraints on measurement locations is shown in Table 5, below. The same set of trial deposits were used to test both the constrained and unconstrained cases.

| | Mined | Abandoned | Total | Accuracy | |
|---|---|---|---|---|---|
| Profitable | **18** | 1 | 19 | 94.7% | |
| Unprofitable | 3 | **78** | 81 | 96.3% | |





| | | | | | |
|---|---|---|---|---|---|
| Total | 21 | 79 | 100 | 96.0% | |
| Profitable Ore | **778** | 36 | 814 | 95.6% | |
| Mean Measures | 9.6 | 5.6 | 6.5 | – | |

**Table 5: Single Body, variable location POMCPOW results with action constraints.**

For the deposits tested, 19% were profitable. Among all the profitable cases, there was a total of 814 units of ore, with POMCPOW deciding to mine 778 units corresponding to 95.6% of profitable ore correctly extracted. On average, POMCPOW took 4.0 more measurements in profitable cases than in unprofitable cases.

We plotted the number of measurements taken before making the final decision in *Figure 21,* below. We also present the average distance from the deposit center in *Figure 22*.

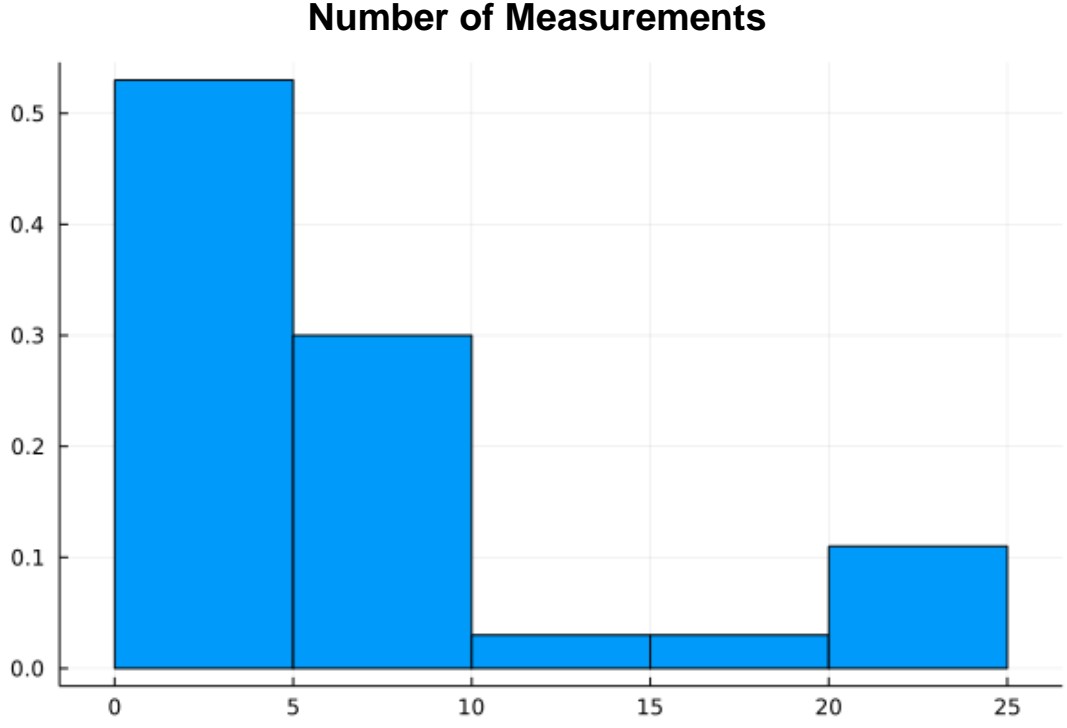

**Figure 21: Measurement histogram, POMCPOW with action constraints, single body with variable location. This figure shows histogram of the number of measurements taken by the POMCPOW solver over all Monte Carlo trials. The trials were limited to a maximum of 25 measurements.**

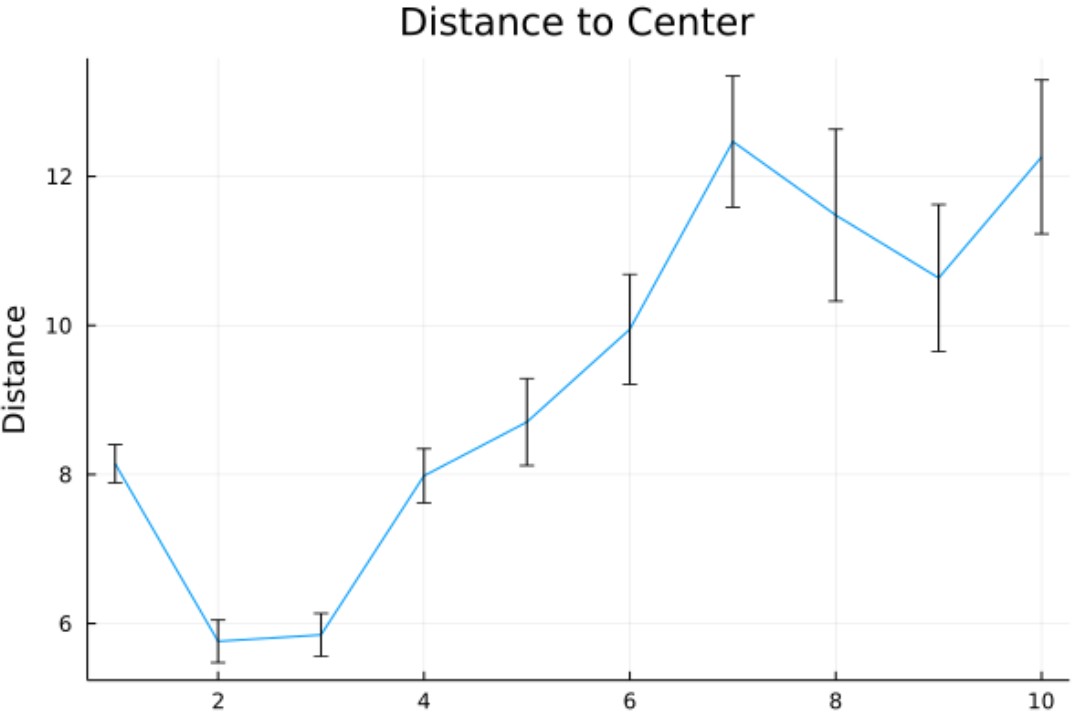

**Figure 22: Measurement distance to center, POMCPOW with action constraints, single body with variable location. The plot shows the average distance between the measurement location and the mineralization center for the measurements at each time step. One standard error bars are also presented.**

### 5.3.2 POMCPOW, Unconstrained Actions

The final decision results for the POMCPOW solver with no constraints on measurement locations is shown in *Table 6,* below.

|  | Mined | Abandoned | Total | Accuracy |
|---|---|---|---|---|
| Profitable | **17** | 2 | 19 | 89.4% |
| Unprofitable | 4 | **77** | 81 | 95.1% |
| Total | 21 | 79 | 100 | 94.0% |
| Profitable Ore | **754** | 60 | 814 | 92.6% |
| Mean Measures | 8.6 | 4.2 | 5.1 | – |

**Table 6: Single Body, variable location POMCPOW results without action constraints.**





625        Among all the profitable cases, there was a total of 814 units of ore, with POMCPOW deciding to mine 754 units corresponding to 92.6% of profitable ore correctly extracted. On average, POMCPOW took 4.4 more measurements in profitable cases than in unprofitable cases.

        As in the constrained test, we plotted the number of measurements taken before making the final decision in *Figure 23*, below. We also present the average distance from the deposit center in *Figure 24*.


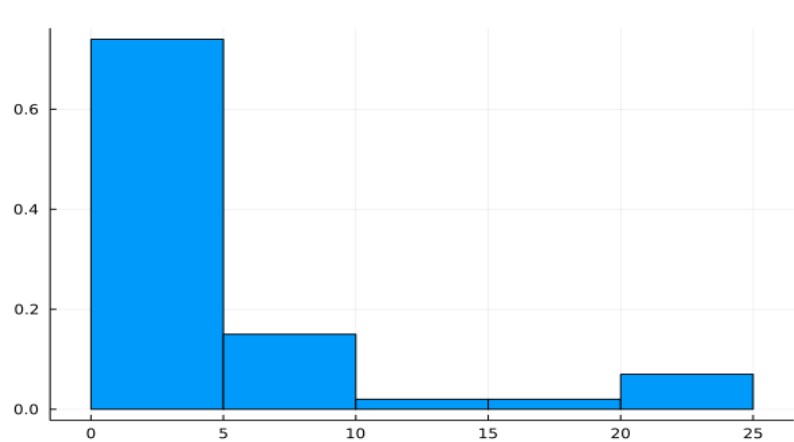

**Figure 23: Measurement histogram, POMCPOW without action constraints, single body with variable location. This figure shows histogram of the number of measurements taken by the POMCPOW solver over all Monte Carlo trials. The trials were limited to**
**a maximum of 25 measurements.**

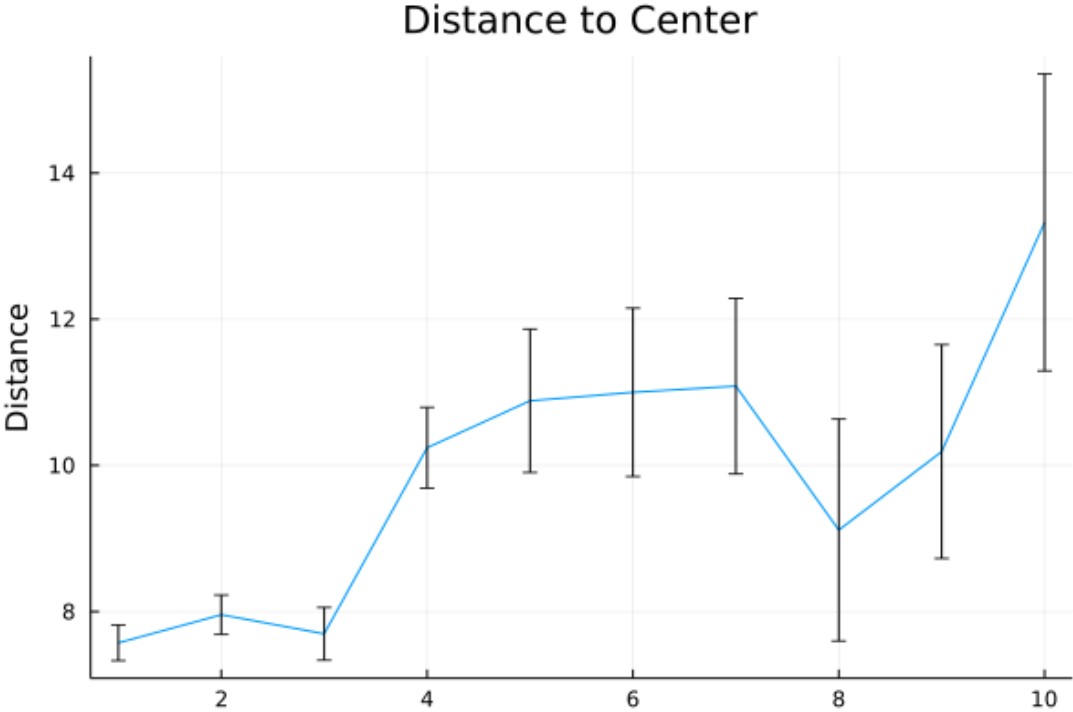

**Figure 24: Measurement distance to center, POMCPOW without action constraints, single body with variable location. The plot shows the average distance between the measurement location and the mineralization center for the measurements at each time step. One standard error bars are also presented.**

## 5.4 Multiple Bodies

In this section, we present the results for the Monte Carlo tests on the case with two mineralization processes located at variable, unknown points in the exploration domain. For every solver, we measured the belief accuracy by calculating the relative mean absolute error (RMAE) of the estimated deposit volume resulting from each measurement. The resulting trends are shown in *Figure 25* with one standard error bounds.



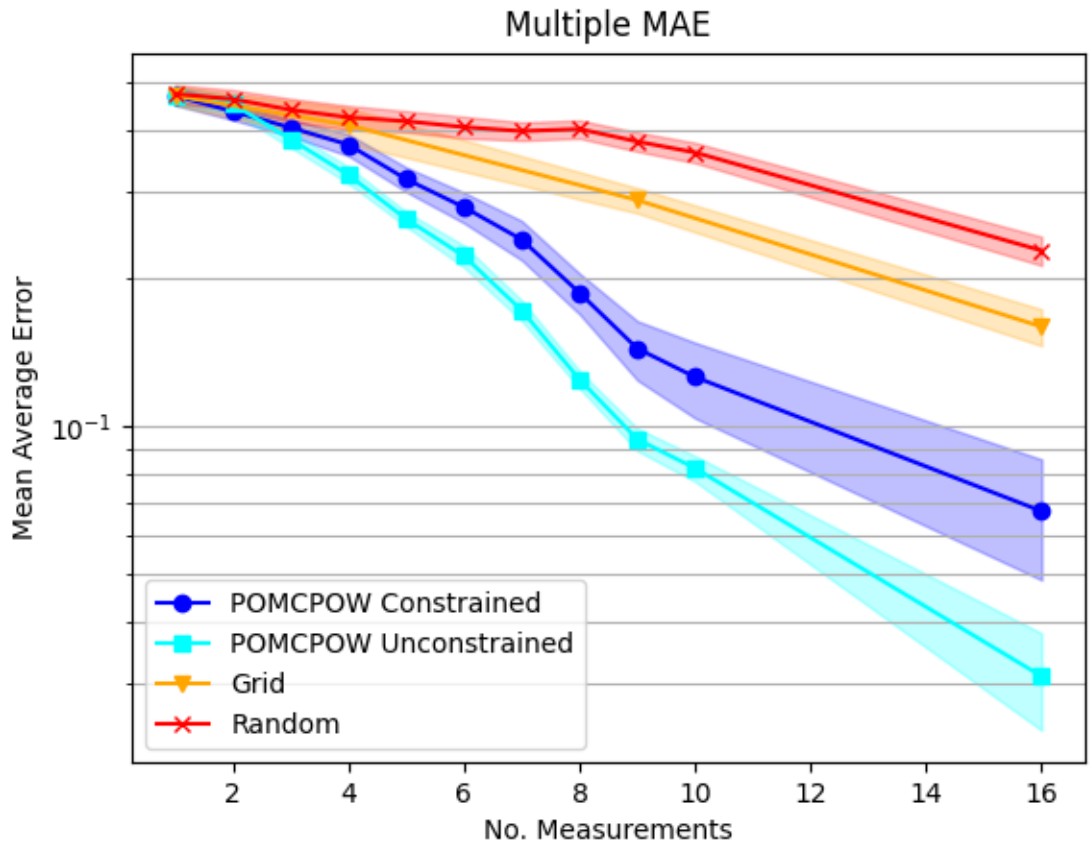

**Figure 25: Relative MAE, two mineralization processes. The plot shows the mean relative absolute error after a given number of measurements taken under each tested method. The mean absolute error is shown along with one standard error bounds for each trend.**

We also measured the change in belief uncertainty by calculating the standard deviation ratios of the belief volume estimate resulting from each measurement. The mean standard deviation ratios over the Monte Carlo trials for each of the solvers is shown in *Figure 26* along with one standard error bounds.



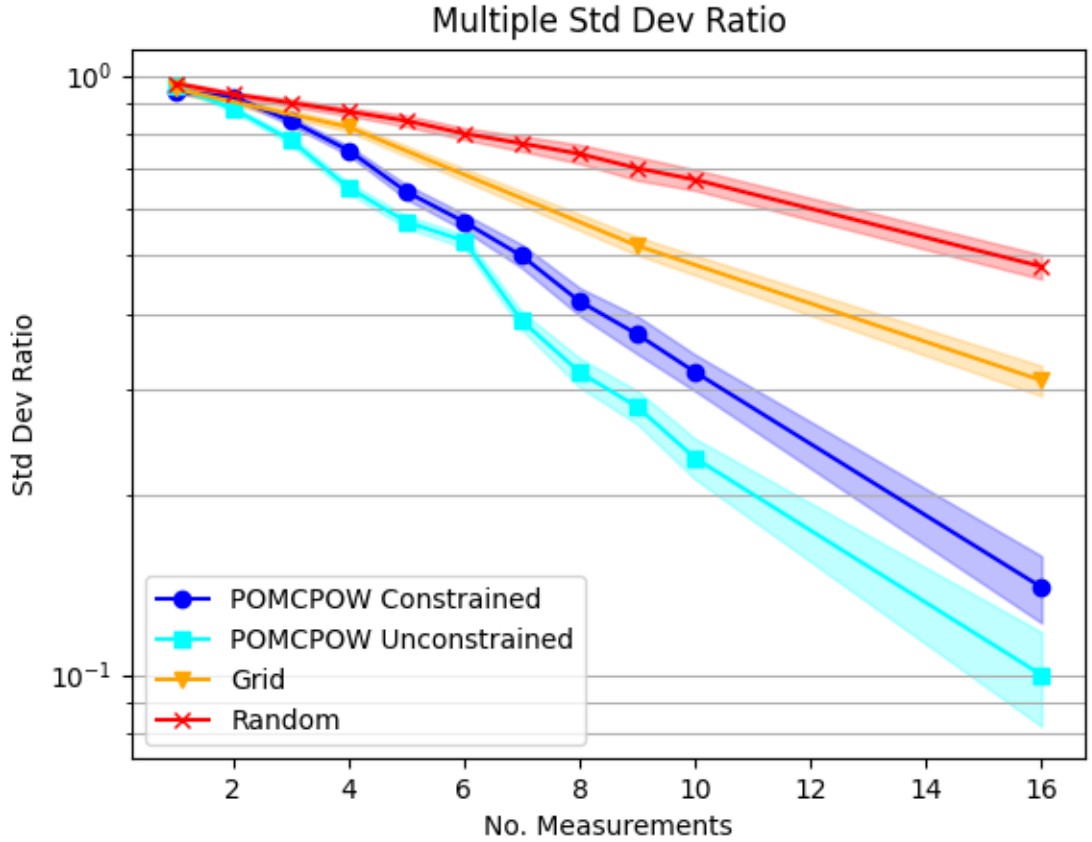

**Figure 26: Two mineralization process standard deviation ratios. The plot shows the mean standard deviation ratio after a given number of measurements taken under each tested method. The mean ratio is shown along with one standard error bounds for each trend.**

### 5.4.1 POMCPOW, Constrained Actions

The final decision results for the POMCPOW solver with no constraints on measurement locations is shown in *Table 5,* below. The same set of trial deposits were used to test both the constrained and unconstrained cases.

|  | Mined | Abandoned | Total | Accuracy |
|---|---|---|---|---|
| Profitable | **13** | 6 | 19 | 68.4% |
| Unprofitable | 1 | **80** | 81 | 98.8% |
| Total | 14 | 86 | 100 | 93.0% |





| | | | | |
|---|---|---|---|---|
| Profitable Ore | **713** | 95 | 808 | 88.2% |
| Mean Measures | 10.1 | 5.4 | 6.2 | – |

**Table 7: Multi-body POMCPOW results with action constraints.**

665        For the deposits tested, 19% were profitable. Among all the profitable cases, there was a total of 808 units of ore, with POMCPOW deciding to mine 713 units corresponding to 88.2% of profitable ore correctly extracted. On average, POMCPOW took 4.7 more measurements in profitable cases than in unprofitable cases.

        We plotted the number of measurements taken before making the final decision in *Figure 27,* below.


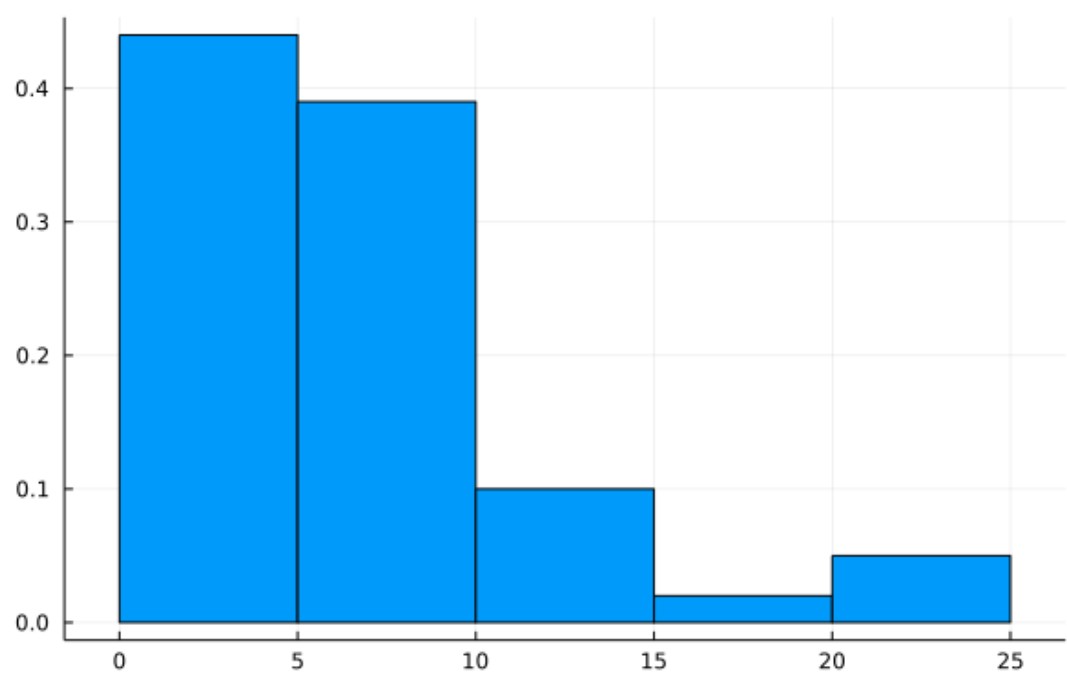

**Figure 27: Measurement histogram, POMCPOW with action constraints, multiple ore-bodies. This figure shows histogram of the number of measurements taken by the POMCPOW solver over all Monte Carlo trials. The trials were limited to a maximum of 25 measurements.**


### 5.4.2 POMCPOW, Unconstrained Actions

        The final decision results for the POMCPOW solver with no constraints on measurement locations is shown in *Table 8,* below.





| | Mined | Abandoned | Total | Accuracy |
|---|---|---|---|---|
| Profitable | **13** | 6 | 19 | 68.4% |
| Unprofitable | 1 | **80** | 81 | 98.8% |
| Total | 14 | 86 | 100 | 93.0% |
| Profitable Ore | **764** | 44 | 808 | 94.6% |
| Mean Measures | 8.9 | 6.1 | 6.5 | – |

**Table 8: Multi-Body POMCPOW results with action constraints.**


    Among all the profitable cases, there was a total of 814 units of ore, with POMCPOW deciding to mine 764 units corresponding to 93.0% of profitable ore correctly extracted. On average, POMCPOW took 3.8 more measurements in profitable cases than in unprofitable cases.

    As in the constrained test, we plotted the number of measurements taken before making the final decision in *Figure*
*28,* below.

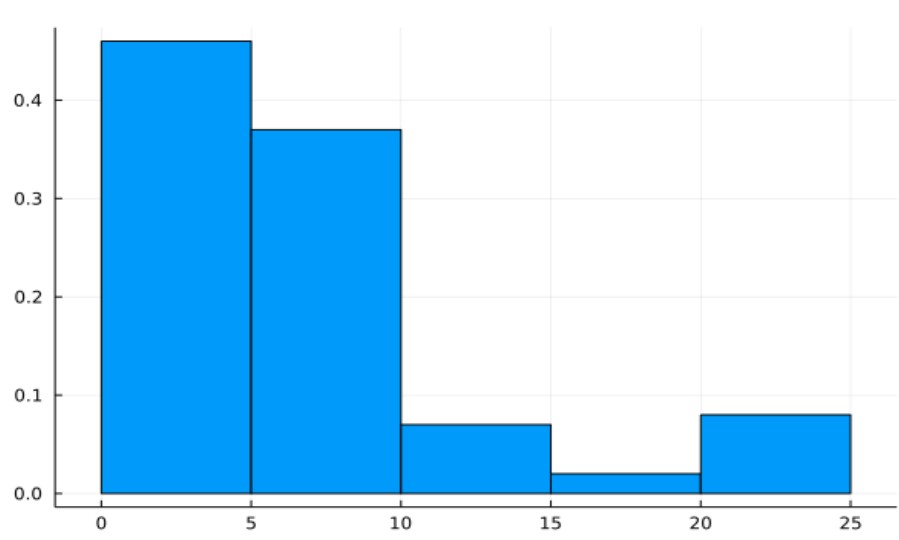

**Figure 28: Measurement histogram, POMCPOW without action constraints, multiple ore-bodies. This figure shows histogram of the number of measurements taken by the POMCPOW solver over all Monte Carlo trials. The trials were limited to a maximum**
**of 25 measurements.**





## 5.5 Deposit Size Sensitivity Studies

The POMCPOW solver was allowed to terminate the measurement campaign at any point before the maximum of 25 measurements were taken. We hypothesized that the size of the deposit being measured would impact how many

measurements POMCPOW decided to take. To test this, we ran POMCPOW on three different deposit sizes for each of the three problem configurations.

1. Sub-Economic: The total massive ore was below the economic cutoff threshold by more than 30% of the threshold value.

2. Borderline-Economic: The total massive ore was within 10% of the economic cutoff threshold value.

3. Economic: The total massive ore was above the economic cutoff threshold by at least 20% of the economic threshold value.

The resulting trajectory of measurements taken by POMCPOW for each of these configurations is shown in *figure 29, figure 30*, and *figure 31* for the single body with fixed location, single body with variable location, and multi-body cases, respectively.




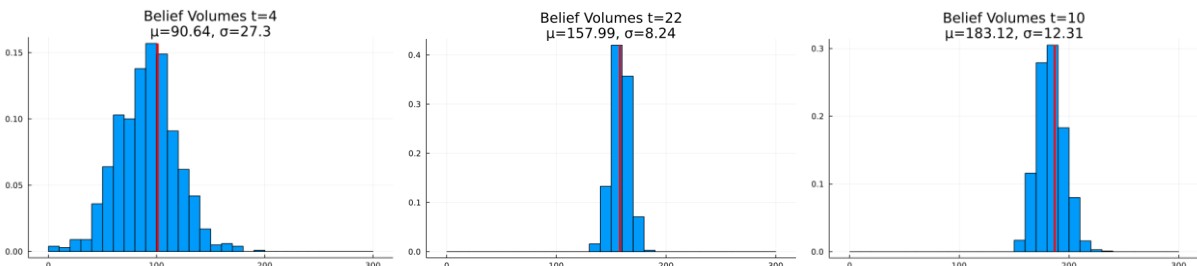

**Figure 29: Deposit size study results for the single body with fixed centroid location case. The sub-economic, borderline, and** 710 **economic cases are shown in the left, center, and right columns, respectively. The top row shows the massive ore present in the tested case. The center row shows the trajectory taken by POMCPOW and the standard deviation of the resultant belief. The bottom row shows the histogram of the ore volumes in the final belief along with the true massive ore volume.**




**Figure 30: Deposit size study results for the single body with variable centroid location case. The sub-economic, borderline, and economic cases are shown in the left, center, and right columns, respectively. The top row shows the massive ore present in the**



tested case. The center row shows the trajectory taken by POMCPOW and the standard deviation of the resultant belief. The
bottom row shows the histogram of the ore volumes in the final belief along with the true massive ore volume.

**Figure 31: Deposit size study results for multi-body case. The sub-economic, borderline, and economic cases are shown in the left,
center, and right columns, respectively. The top row shows the massive ore present in the tested case. The center row shows the
trajectory taken by POMCPOW and the standard deviation of the resultant belief. The bottom row shows the histogram of the ore
volumes in the final belief along with the true massive ore volume.**

730        The number of measurements taken in each tested configuration are summarized in *table 9*. In all three problem

configurations, POMCPOW made significantly fewer measurements on the sub-economic deposits than it did on the

borderline or economic deposits. In the single-body cases, POMCPOW measured the borderline-economic deposits more



than the economic case. In the multi-body case, POMCPOW reached the maximum of 25 measurements for both the borderline, and economic cases.


|  | Sub-Economic | Borderline | Economic |
|---|---|---|---|
| Single-Body, Fixed Location | 4 | 22 | 10 |
| Single-Body, Variable Location | 5 | **25** | 23 |
| Multi-Body | 13 | **25** | **25** |

**Table 9: Deposit size study summary. The total number of measurements taken by POMCPOW before terminating the measurement campaign is shown in for each test configuration and deposit size. Cases in which the maximum 25 measurements were taken are shown in bold.**

We examined the results of the Monte Carlo studies for a trend in the measurement campaign length. There was a positive correlation between the size of the mineral deposit and the number of measurements taken in the single-body cases. This trend is shown in Figure 32. The multi-body cases did not have a significant number of trials with fewer than ten measurements.

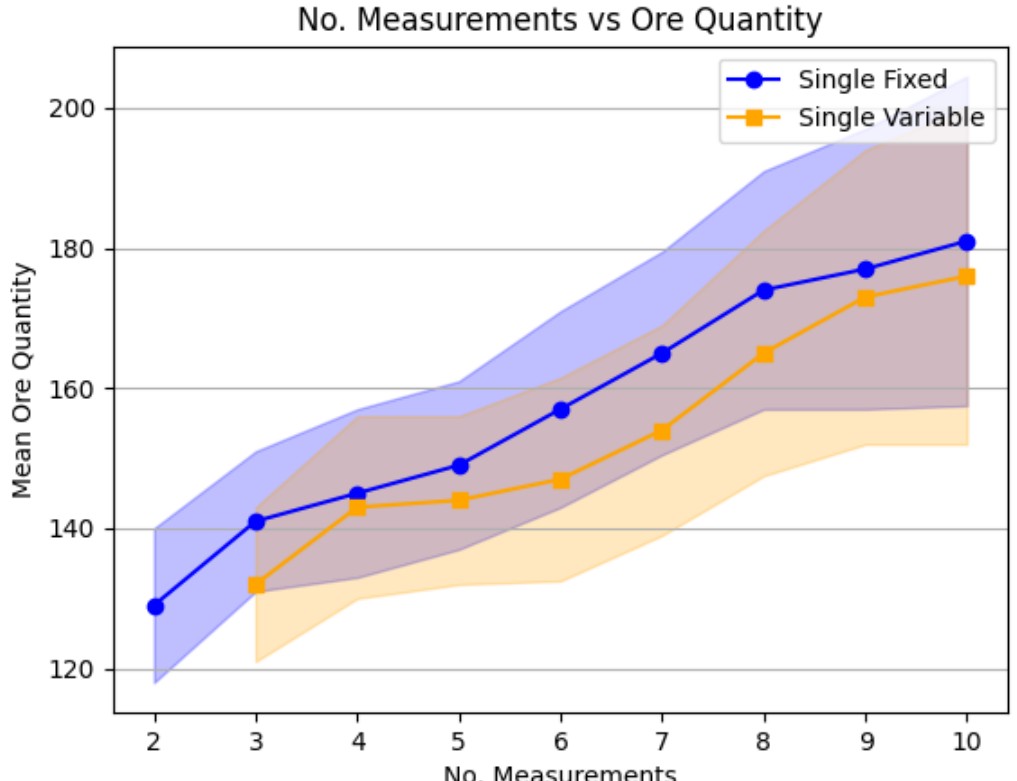

**Figure 32: Measurement campaign length and deposit size. The mean deposit size is shown for different measurement campaign lengths, along with one standard-error bounds.**





## 6 Discussion

In all three deposit configurations tested in the Monte Carlo studies, the measurements taken by POMCPOW tended
to improve the RMAE and the standard deviation ratio of the resulting belief significantly more quickly than the grid pattern
and the random methods. In all cases, POMCPOW tended to reach the accuracy and precision of the full sixteen
measurement grid after just seven to ten measurements. With increasing complexity of the problem (more uncertainty, more
bodies) the difference in performance between the AI and the grid pattern method increases.

In the single-body cases, the performance of the POMCPOW solver with and without action constraints was not
generally significantly different. In several cases, the constrained trajectories outperformed the unconstrained trajectories in
terms of both belief accuracy and variance. This suggests the POMCPOW solver did not completely converge in the
unconstrained cases, since the constrained trajectories are necessarily a subset of those reachable in the unconstrained case.
This is likely a result of the unconstrained problem having significantly more locations for POMCPOW to select from at
each step. Converging on larger search spaces tends to require more trial simulations in POMCPOW to converge. In the
presented experiments, the POMCPOW trials were run with the same number of rollouts in both the constrained and
unconstrained cases. In the multi-body cases, the unconstrained solver did tend to outperform the constrained solution. This
suggests that the constraints pose a more significant limitation to the solution in the multi-body case than in the single-body
case.

In the single-body cases, the final MINE or ABANDON decisions made by POMCPOW were accurate in both
economic and non-economic cases, choosing the correct decision in over 90% of cases in most test configurations. The
accuracy in non-economic cases tended to be slightly higher than in economic cases. This is likely the result of sub-
economic deposits being more common in the prior distribution than economic deposits, and the initial belief expected ore
volume starting below the economic threshold. The percentage of profitable ore mined tended to be higher than the ratio of
correct mining decisions. For example, in the single-body fixed location case with measurement constraints, POMCPOW
correctly identified approximately 89% of the profitable cases, though it mined 95% of all the profitable ore. This suggests
that the economic cases which POMCOW failed to correctly identify were only marginally economic.

The accuracy of the final POMCPOW decisions decreased significantly in the multi-body cases. In approximately 32%
of profitable cases, the algorithm incorrectly decided to abandon the prospect. Inspection of the test results suggested that
this was due to the belief model (Bayes model) failing to correctly resolve one of the two ore bodies before making a
decision. An example of this is shown in *Figure 32*, where the algorithm incorrectly abandoned the marginally economic
deposit after seven measurements before resolving both bodies. This behavior is likely caused by the belief incorrectly
concentrating probability on a sub-economic, single body cases, not by the POMCPOW algorithm. The observed belief
behavior was likely a result of the particle ensemble failing to retain a sufficient number of multi-body instances. Many





methods have been proposed to monitor and prevent this type of particle filter degeneracy (Thrun, 2005), hence, future
research will focus on including better particle filter methods for these types of problems

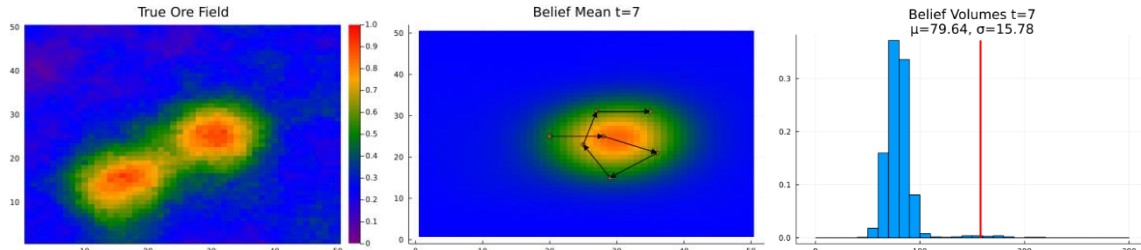

**Figure 33: Multi-Body Failure Example. This figure shows an example of an incorrect ABANDON decision made on the multi-body case. In this trial, the belief converged too quickly to a sub-economic case with a single ore-body before resolving the second**
**ore body in the south west.**

Interesting emergent behavior was observed in the single-body cases. The initial measurement was not typically taken at the center of the belief distribution but was instead offset slightly. The subsequent measurements tended to step-in towards the center before gradually moving outward. This behavior can be understood as intuitive extent-finding methodology. Each
measurement is taken to try to locate the edge of the deposit, where the most information about the deposit size can be learned. As more information is gained near the center, where positive observations are more likely, the measurements tend to move outward toward more informative, but higher variance data may be gathered.

One important feature of the defined POMDP is that it allows the solver to make a variable number of measurements before concluding. In each case studied, a wide variety of trajectory lengths were observed. Because there is a
cost per-measurement and a time discount on the eventual reward, POMCPOW tended to prefer shorter measurement campaigns, when possible, with fewer than five measurements being the mode in most cases. However, clear evidence of truncation at the upper end can be seen in the measurement histograms, suggesting that in some cases, more than the maximum allowed 25 measurements would have been taken had the limit not been imposed. In general, it was observed that POMCPOW took more measurements on cases that we would consider more difficult. On cases that were borderline
economic, in which resolving the deposit size with good fidelity was necessary to make the correct final decision, POMCPOW tended to take more measurements. For clearly sub-economic cases, POMCPOW abandoned after just a few measurements. For clearly economic cases, POMCPOW took more measurements than in clearly sub-economic cases. This is likely caused by the initial belief starting with an expected sub-economic value. This would require more Bayesian updates to converge toward an economic value than a sub-economic value. We also noted that fewer measurements were
taken in the fixed-location cases than in the variable location cases. This is likely the result of the latter cases requiring the POMCPOW solver to localize the deposit in addition to measuring its extent.





# 7 Conclusion

In this work, we presented a Bayesian sequential decision-making approach to improving geoscientific model through sequential data acquisition planning, with application to mineral exploration. We presented a framework to model challenges like mineral exploration problems by means of partially observable Markov decision processes (POMDPs). We demonstrated the general method with a specific example case in which we solved a 2D mineral exploration problem with a known exploration area. To solve this problem, we developed a hierarchical Bayesian belief using a particle filter and Gaussian process regression and the Monte Carlo search algorithm POMCPOW.

The results of our studies demonstrate that a closed-loop sequential decision-making approach significantly outperforms a typical fixed-pattern grid approach. The measurements recommended by POMCPOW improved the accuracy and variance of the belief over the deposit extent significantly faster than the baseline methods. The resulting behavior that emerged from POMCPOW was intuitive and tended to result in shorter measurement campaigns than a fixed pattern resulting in comparable accuracy.

The methods presented in this work are general to many areas of resource exploration. The belief and solver presented for the test case are not necessarily required to implement this approach. Future work should apply these methods to higher fidelity exploration problems using more realistic geological models and measurement simulations, such as geophysical surveys. The POMCPOW solver was chosen because it is generally applicable to many POMDPs without modification. However, as seen in the unconstrained cases, POMCPOW may have not converged to an approximately optimal solution. Future work should investigate modifications to the baseline POMCPOW algorithm to improve its performance in exploration tasks. Extensions to POMCOW should be explored to use the fact that the deposit state underlying the belief is static to reduce the variance of the value estimates and the required sample complexity of the search. Future work should also investigate other solver types, such as point-based value iteration (PBVI), that may handle high-variance beliefs more efficiently.

## Code/data availability

The current version of Intelligent Prospector is available from the project website: https://github.com/sisl/MineralExploration under the MIT License. The exact version of the model used to produce the results used in this paper is archived on Zenodo (Mern, 2022 10.5281/zenodo.6727378), as are input data and scripts to run the model and produce the plots for all the simulations presented in this paper (Mern, 2022 10.5281/zenodo.6727378).





**Author contribution**

John Mern developed the code, methodologies and conceptualization

Jef Caers developed methodologies and conceptualization as well as project supervision

Both equally contributed to the writing


**Competing interests**

The authors declare that they have no conflict of interest.

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
