# Peer review of "Intelligent prospector v1.0: geoscientific model development and prediction by sequential data acquisition planning with application to mineral exploration"

_Geoscientific Model Development, 2022_

## Author Comment (AC1)

*I enjoyed reading this paper! It is a nice piece of work on a very interesting topic.*

Many thanks for your encouragement!

*My main comments are related to i) comparison of algorithmic parameters and ii) extensions to 3D. Comment i) requires some work, but I think it should be relatively fast to do in a revision. Comment ii) can be discussed some more and left for future work.*

*i) Your paper contains a number of cases, but there are limited comparison of the suggested method using different tuning parameters m, k, alpha. I am guessing by tuning some of these one could have a greedy approach at one end versus a very deep one which is more time consuming at the other end. But I don't see much comparison of using various of these (extreme) inputs as it is now. I am also not sure how easy it would be to compare the suggested approach with ones like Q-learning or other RL / value iteration methods for your case?*

The computational expense of the algorithm is primarily controlled by the number of trial trajectories generated $m$. In general, higher $m$ will result in larger trees with deeper branches and higher computational cost. Changing progressive widening parameters ($k$ $a$) can also change the computational expense and depth of search (and therefore the greediness of the resultant policy). Overly aggressive widening will tend to result in short-sighted policies that are one-step greedy, since the Monte Carlo estimates for each action will tend to be dominated by very short horizon trajectories. In our problem, this would tend to result in the degenerate behavior of always abandoning the prospect on the first step, since that was the only action with a non-negative expected one-step return.

Reinforcement learning based approaches may also be used to solve the presented mineral exploration POMDP. Without augmentation, they are likely not as well suited as the presented Monte Carlo method. Reinforcement learning methods such as Q learning or policy gradient learning, learn *offline* policies before any actions are taken in the real world. This requires functions to represent policies with functions that can generalize training examples to new experiences. Because these methods learn policies for the entire space of experiences that may be encountered, they tend to require significantly more training data than an online method, such as POMCPOW, that only solves for a single problem being encountered.

Reinforcement learning methods are also formulated for fully-observable domains, without explicitly accounting for state uncertainty. This can be addressed in several ways to allow RL methods to function, however, they are not as efficient as methods developed for domains with state uncertainty. In particular, RL methods tend to use very basic random sampling for exploration. Research has shown that UCB-type exploration (upper confidence

bound) in a tree is significantly more efficient and can make a large impact on problems where information gathering is important.

Value iteration methods may also be used here, with approximations for the continuous state, action, and observation spaces. Approximate value iteration methods, like point-based value iteration, do not allocate computation time as efficiently as MCTS like methods, as the tree tends to be constructed prior to learning, so that learned experiences cannot inform where to grow the tree most efficiently.

We added this additional material in the introduction and discussion section

*ii) In practical mining operations, wouldn't there ordinarily be sequential 3D boreholes where one can choose and modify the drilling order / locations? One could also potentially stop data collection (and drilling) in one borehole after a certain depth (before the initial planned depth is reached). Along boreholes one could also have different data collecting frequency. The suggested strategy for collecting data seems a bit restricting in this setting - as it is 2D only in this paper. What more is needed or possible in 3D?*

Yes, these are indeed various options. In 3D we would consider

- Location, orientation (azimuth, dip) and depth. All these can be made variable in the approach, but doing so would require extending the problem to continuous parameters. We have added a paragraph in the discussion that addresses the extension to 3D and what extra that would require
- POMCPOW can handle continuous actions as it is currently implemented.Increasing the number of parameters needed to define an action (adding azimuth + dip + depth to the current x, y) tends to increase the amount of computation needed to solve the problem. In such cases, a problem-specific policy can be used to augment the basic UCB exploration method.

*iii) Some detail comments:*

*- Mark a_1 and a_2 on first axis of Figure 1, as well as have 'x' or similar as the axis label.*

    Agreed

*- l175: This is accounts?*

    Typo: correct to "This  accounts for…"

*-Around Table 1, I don't think all these comparison of AI and geo terminology are needed.*

We have found it very useful in our own collaboration, so we believe it may in fact be needed to bridge fields

*-In Sect 4.1 there is a discussion of "actions", and you state 'the agent may acquire measurements (data)'. But at this point in the presentation there is no observation terminology 'o'. Aren't the action here to mine or abandon?*

Line 196. Should we include an additional sentence "The agent may also decide to abandon or proceed to mine the prospect"?

*-Not sure $L(o_{t+1}|...)$ is defnied in l 225 expression (it comes much later, I think)?*

This is defined on line 214 as $Z(o...)$, we should change it there or at line 225 to be consistent

*-Algorithm 1, data line should have d <- d + e, e \sim N(0,\sigma_n^2)*

Agreed

*-Sunberg and Kochenderfer, 2018 paper is not on the reference list?*

Agreed

*- \sigma means several different things in the paper and can be a bit confusing.*

Agreed

*-You often say Figure X below. You don't need the 'below' here.*

Agreed

*-Would it be possible to color-code the histograms in Fig. 15 (+ similar ones) according to 'mine' or 'abandon' ? Couldn't you also have one bar for each outcome here, rather than bins 0-5, 5-10, etc.?*

That is changed

**Reviewer 2**

*Thanks for your interesting contribution. The manuscript is well written and very pleasant to read. The objectives are very clear and the method is rather well explained. I have a few suggestions and questions to clarify some points and facilitate the reproducibility of the work.*

Thank you for your encouragement!

Line 56: could you explain what a non-sequential scheme could be in the context of mineral exploration, as it seems to contradict the previous sentence on line 38. It becomes clearer though, when reading the following paragraphs.

Last paragraph of section 1: Which of the mentioned approaches did you select for your demonstration?

Monte Carlo Planning

Section 4.2: how is the state space initialized?

We are a bit confused with this questopn. A state space is not initialized as it is just a space in the mathematical definition. I think the question may be asking how the space is defined/parameterized. In that case, I would say the space is defined by a function and set of parameters that define a massive anomaly (e.g. the center and radius of a circle), and a sample of a Gaussian random field from a Gaussian process. In practice, we represent the combination of these two as a two dimensional array, where each cell of the array represents the mineralization at that location.

Line 289: should it be r(s,a)= -Cost(s,a) or Cost(s,a)=-Cmeasurements to be consistent with the substraction of Cextraction in the profit?

We agree. I prefer the r = - cost representation for clarity.

Table 2: where does d come from? Formatting: should it be an algorithm rather than a table object? See e.g. the example in the latex template

d is the value of a particular particle in the ensemble.

Line 356: 'At each time step'

Line 357: 'The full tree is constructed' – in the case of the POMDP ?

Yes we can see why this is confusing. The "full" tree here does not refer to the complete tree possible under a POMDP. A better wording of what we're trying to communicate here may be "The tree construction process is completed before …"

Line 359: by trial trajectory, do you mean a branch of the tree or realization of the full tree? How is the (partial) tree generated? What is the prior over the trajectory length , and between the different actions (explore further, mine or abandon)

A trial trajectory is a simulation that starts at the root node and continues until a termination condition is reached (e.g. maximum measurements have been taken or "MINE/ABNANDON" was selected). A portion of each trial trajectory is added to the existing tree structure every step. The logic of how the tree is explored and constructed is defined in the POMCPOW paper. Fully explaining the logic here would require a more in-depth description than is in the scope of this paper.

Line 378: previous visits cumulated over the previous iterations t ?

Agreed

Lines 423 to 425 and figure 9 bottom right panel: can you clarify the stopping criteria as at =5 the mean of the ensemble decreases and is getting smaller than the extraction cost. Can you also clarify how the value of gained information is assessed?

We are not sure we understand the first part of this question. The stopping criteria is not explicitly defined, beyond a maximum number of measurements allowed. The stopping behavior of the agent is learned during the optimization process.

The value of information is not explicitly calculated. An optimal policy would continue to gather information by taking measurements, so long as the value of that information exceeds the measurement's cost. The value of the information would be calculated by the difference between the expected value of the "MINE/ABANDON" decision with and without the information.

Figure 13: missing scale for the mean average error

Thanks for your careful reading

---

## Author Response (AR2)

**Reply letter, reviewer #2**

Thanks to the authors for clarifying their manuscript.

I have some additional minor points for clarification, that I unfortunately did not notice during the first review round. It should help the reader who want to reproduce your work or apply your method.

Figure 1: A, B, C, D called in text and caption but missing from the figure, though obvious.

corrected

Line 147: though it does not affect the methodology presented in the paper, wouldn't it make more sense to define i(x) as the indicator of m(x)>t rather than z(x)>t ? It would also further support inferring m(x), unless the prior of m(x) ad r(x) is better characterised than the prior of z(x).

Indeed both options can be made

- i(x) as indicator of m(x)
- i(x) as indicator of z(x)

In reality the r(x) is more complex, as noted in the text and consisting of "confounding elements" related to any modification of the ore body, so in a real context r(x) will not be a simple additive noise term.

Change, line 117: "…and hence the noise term in this simple example is used to develop a methodology"

Line 159: based on a minimum estimated volume of ore?

Changed: "The question we will address is: what is the optimal sequence of data acquisition that best informs "mine" vs "do not mine" decision, based on a mineable volume exceeding some minimum threshold?"

Line 226: I am a bit confused by b' – do you mean b(s_{t+1}) proportional to L(o_{t+1} given s_{t+1} and a_t) times b(s_t) ? b' is only used later in the pseudo-algorithm

This a common notation used in AI

The b' notation here is the posterior so it includes the condition | o_{t+1}

Change line 227: "Note that $b'(s_{t+1})$ is AI notation for a posterior $p(s|o)$, where $p(s)$ is the prior"

Figure 4: use t and t+1 rather than t-1 and t to be consistent with the caption and manuscript description?

Corrected

Line 333: it looks like r(x) is missing – do you mean f(m,r , o)=f(o)*f(m given o)*f(r given m) ?

Lines 335 to 337: I am confused by the notation versus the description of f(m given o) (conditional belief in my understanding) and f(m,o) (joint belief in my understanding), can you check this?

Line 342: posterior f(m given o) ?

Line 349 and 351: I am not sure to understand how the decomposition of o_t as o_tm+o_tr and the determination of o_tm by m(x) lead to these weight equations.

We apologize for this issue; it seems something happened when converting to Word from google doc.

Here is the section with correct equations

$$o_{1:t} = \{o(x_\alpha), x_\alpha = 1, \dots t\}$$

The observed measurements are dependent upon both random functions, $m(x)$ and $r(x)$, hence a traditional conditional simulation cannot be directly applied. Instead, we formulate this problem as a hierarchical Bayes' problem by factoring the joint distribution into

$$f(m(x), r(x)|o_{1:t}) = f(m(x)|o_{1:t}) \times f(r(x)|m(x), o_{1:t})$$

Samples are generated from this distribution hierarchically by first drawing a sample from the distribution over $m(x)$

and then using the resulting sample to draw from the conditional distribution over $r(x)$. We model the belief $f(m(x)|o_{1:t})$ as a particle set and update it using an importance resampling particle filter (Del Moral 1996, Liu et al. 1997). The conditional belief $f(r(x)|m(x), o_{1:t})$ is modeled as a conditional Gaussian process.

A particle set is an ensemble of realizations of the state variable with a sample distribution approximating the true state distribution. The initial particle set is generated by first sampling an ensemble from the uniform prior distribution. For an

$n$ particle set, this corresponds to an ensemble of $\left(m^i(x), r^i(x)\right), i = 1, \dots n$ where each particle is equiprobable.

When new information $o_t$ is observed, the particle filter updates the belief by updating the ensemble such that the new particles are sampled according to the posterior distribution $f(m(x)|o_{1:t})$. To do this, a posterior weight is calculated for each particle according to Bayes' rule as

$$w^i \propto f(o_t|m(x), o_{1:t-1})$$

Note that each particle is treated as equiprobable in the particle set, so the prior probability is dropped in the above expression.

The observed measurement $o_t$ is determined by the sum of $m(x)$ and $r(x)$ at the location of the measurement. We denote these values as $o_t^m$ and $o_t^r$, respectively, such that $o_t = o_t^m + o_t^r$. Using this notation, we can decompose the particle weight function into

$$w^i \propto f\left(o_m^t|m(x)\right) \times f(o_r^t|m(x), o_{1:t-1})$$

Because the value of $o_m^t$ is completely determined by $m(x)$, we can simplify this further to

$$w^i \propto f(o_t - o_m^t|o_{1:t-1} - m(x))$$

Line 370: insert 'At' before 'each time step t'

corrected

Line 422: do you mean n (size of the particle set) = 1E4 simulations? It becomes clear only in the conclusions that you refer to the total number of trial trajectories m. What is n, the size of the particle set?

What is meant is "trial simulations" or trial trajectories related to evaluation of the Monte Carlo Tree search,

Change, line 422: "We ran POMCPOW for 10,000 trial simulations (trajectories) per-step"

Did you define a maximum number of time steps 1) to restrain the depth of the search tree and 2) in the iterative process?

We define 25 measurements as a maximum but allow POMCPOW to search over full depth (selectively deepening search process)

Change, line 457: "We limited the agent to a maximum of 25 measurements"

Figure 16: would it be worth to plot the approximate radius of the orebody as an horizontal indicative dashed line?

Very nice suggestion, we changed all figures like these, here is an example

[Figure]

Here the dotted line is the maximum possible orebody. The fact that we step out further is because of the imperfect measurements we take on it